# Characterization of two *O*-methyltransferases involved in the biosynthesis of *O*-methylated catechins in tea plant

Ji-Qiang Jin [1,4], Fu-Rong Qu[1,4], Huisi Huang[2,4], Qing-Shuai Liu[1,4], Meng-Yuan Wei[1,4], Yuee Zhou[2], Ke-Lin Huang[3], Zhibo Cui [2], Jie-Dan Chen[1], Wei-Dong Dai[1], Li Zhu[1], Ming-Zhe Yao [1] ✉, Zhi-Min Zhang [2] ✉ & Liang Chen [1] ✉

Tea is known for having a high catechin content, with the main component being (−)-epigallocatechin gallate (EGCG), which has significant bioactivities, including potential anti-cancer and anti-inflammatory activity. The poor intestinal stability and permeability of EGCG, however, undermine these health-improving benefits. *O*-methylated EGCG derivatives, found in a few tea cultivars in low levels, have attracted considerable interest due to their increased bioavailability. Here, we identify two *O*-methyltransferases from tea plant: CsFAOMT1 that has a specific *O*-methyltransferase activity on the 3″-position of EGCG to generate EGCG3″Me, and CsFAOMT2 that predominantly catalyzes the formation of EGCG4″Me. In different tea tissues and germplasms, the transcript levels of *CsFAOMT1* and *CsFAOMT2* are strongly correlated with the amounts of EGCG3′Me and EGCG4″Me, respectively. Furthermore, the crystal structures of CsFAOMT1 and CsFAOMT2 reveal the key residues necessary for 3″- and 4″-*O*-methylation. These findings may provide guidance for the future development of tea cultivars with high *O*-methylated catechin content.

Tea is one of the most consumed beverages in the world. The popularity of tea lies in its unique flavor and health features, conferred by the diverse bioactive compounds in tea leaves, such as caffeine and catechins (flavan-3-ols)[1]. Tea catechins, which constitute more than 12% of the dry weight, are a class of bioactive polyphenols, with (−)-epigallocatechin-3-gallate (EGCG) being the most abundant ingredient[2]. Extensive studies have demonstrated various health-promoting properties of EGCG, including anti-cancer[3,4], anti-inflammatory[5], and cardioprotective effects[6], where EGCG acts as a

potent antioxidant or inhibitor of DNA methyltransferases and several signaling pathways[6,7]. However, the therapeutic efficacy of EGCG is dramatically diminished due to its instability in the neutral to basic conditions of the intestine, which causes oxidation of the EGCG[7]. Moreover, the highly polar nature of the polyphenolic structure makes it difficult to permeate the intestinal tract wall, further limiting its bioavailability in the human body.

 *O*-methylated EGCG refers to a series of methylated derivatives formed by modifying the phenolic hydroxyl groups on the galloyl

[1]Key Laboratory of Biology, Genetics and Breeding of Special Economic Animals and Plants, Ministry of Agriculture and Rural Affairs; Tea Research Institute of the Chinese Academy of Agricultural Sciences, Hangzhou 310008, China. [2]College of Pharmacy, Jinan University, Guangzhou 510632, China. [3]State Key Laboratory of Tea Plant Biology and Utilization, Anhui Agricultural University, Hefei 230036, China. [4]These authors contributed equally: Ji-Qiang Jin, Fu-Rong Qu, Huisi Huang, Qing-Shuai Liu, Meng-Yuan Wei. ✉e-mail: yaomz@tricaas.com; 1363210756@163.com; liangchen@tricaas.com

moiety of EGCG with methyl groups (Fig. 1a). In tea plants, natural methylation occurs mainly in the young leaves, specifically at the 3″- and 4″-positions of EGCG, resulting in the generation of EGCG3″Me ((−)-epigallocatechin-3-*O*-(3-*O*-methyl)-gallate) and EGCG4″Me ((−)-epigallocatechin-3-*O*-(4-*O*-methyl)-gallate), respectively[8,9]. In a previous study, we investigated the content of EGCG3″Me and EGCG4″ Me in 27 accessions of tea germplasms with different genetic backgrounds and detected EGCG3″Me in about half of the accessions, while EGCG4″Me was detected only in three accessions[10]. It has been reported that the *O*-methylated EGCGs have better stability, lipid solubility, higher oral bioavailability, and stronger biological activity compared to EGCG[11]. Plasma concentration profiles of EGCG and *O*-methylated EGCGs in rats revealed a sevenfold higher concentration of EGCG3″Me in plasma after oral administration than EGCG and EGCG4″ Me[12]. Clinical trials have also demonstrated that a tea with *O*-methylated EGCGs ('Benifuuki') has stronger antiallergy[13] and antihypertensive[14] effects than a tea without *O*-methylated EGCGs ('Yabukita').

The remarkable bioactivity of EGCG3″Me has aroused great interest in growing tea plants that are rich in this compound. Despite the abundance of tea[2], tea plants naturally rich in *O*-methylated EGCGs (>10 mg g$^{-1}$) are scarce[15]. In addition, tea plants typically undergo a long juvenile stage in their life cycle, making it challenging and time-consuming to develop new cultivars through conventional breeding methods or genetic modification. Another hurdle in the genetic modification of tea plants is the limited understanding of the molecular mechanisms underlying the *O*-methylation of EGCG. Previous studies have shown that *O*-methyltransferase (OMT) can catalyze the formation of *O*-methylated EGCG (Fig. 1a)[16], and a caffeoyl-CoA-*O*-methyltransferase (CCoAOMT) has been successfully cloned from tea plant[17]. However, in vitro enzymatic reactions with CCoAOMT yielded a variety of *O*-methylated EGCG products, including EGCG3″Me, EGCG4″ Me, (−)-epigallocatechin-3-*O*-(3,5-*O*-dimethyl)-gallate and (−)−3-*O*-methyl-epigallocatechin-3-*O*-(3,5-*O*-dimethyl)-gallate. This finding contradicts the fact that the major *O*-methylated EGCGs detected in tea plants are EGCG3″Me and EGCG4″Me, suggesting that the genuine OMTs responsible for *O*-methylation at the 3″- and 4″-positions have yet to be identified.

In this study, we identify two *O*-methyltransferases (CsFAOMT1 and CsFAOMT2) from a tea plant. CsFAOMT1 exhibits a specific *O*-methyltransferase activity at the 3″- position of EGCG, and the level of EGCG3″ Me in different tissues and tea plants with different genetic backgrounds is closely correlated with the transcript level of *CsFAOMT1*; whereas CsFAOMT2 can methylate EGCG at both the 3″- and 4″-positions, with EGCG4″Me being the dominant product. The crystal structures of CsFAOMT1 and CsFAOMT2, in combination with in vitro enzymatic studies, revealed the key residues essential for 3″- and 4″-*O*-methylation. These results may guide the future breeding of tea plants with high *O*-methylated EGCGs and the development of new tea products.

## Results

### Cloning candidate *O*-methyltransferase genes from the tea plant
We first aimed to identify the key gene responsible for EGCG3″Me biosynthesis in 'Jinxuan' (JX) and 'Zijuan' (ZJ), two famous Chinese tea cultivars containing EGCG3″Me. To achieve this, we performed bulked segregant RNA sequencing (BSR-Seq) on the F$_1$ population by crossing JX and ZJ[18]. A total of 24 individuals with high EGCG3″Me content (>5 mg g$^{-1}$) and 24 individuals with low EGCG3″Me content (<0.1 mg g$^{-1}$) in the F$_1$ population were selected to form the two extreme groups (group H/L). BSR-Seq analysis was performed on an Illumina NovaSeq platform, and approximately 250,000 single nucleotide polymorphisms (SNPs) or insertions/deletions (InDels) were identified. We found that the variants associated with the trait were enriched in a region of about 46 Mb on Chromosome 6 (GWHAZTZ00000004) of the 'Huangdan' reference genome[19],

suggesting that the target gene may be localized in this region (Fig. 1b). In addition, a total of 217 differentially expressed genes (DEGs) were identified, including 94 upregulated DEGs and 123 downregulated DEGs. Among the 94 upregulated DEGs (Supplementary Data 1), five DEGs (HD.03G0010060, HD.03G0009980, HD.03G0009970, HD.03G0009860, HD.06002937) were putative *OMTs* that were located in the region identified on Chromosome 6. To pin down the target *OMT*, 27 accessions of tea germplasms with distinctly different genetic backgrounds were subjected to RNA-seq to analyze the correlation between the transcriptional level of 35,628 tea plant genes, including the five *OMTs*, and the level of EGCG3″Me (Supplementary Data 2). Ultimately, we found that tea plants expressing higher levels of *OMT* (HD.03G0010060) tended to contain more EGCG3″Me, indicating a positive correlation between them ($R = 0.89$, $P = 2.75 \times 10^{-28}$).

To elucidate the sequence of the candidate *OMT*, we designed primers and cloned the coding region from the cDNA of JX, which encoded a protein of 236 amino acids (Supplementary Figure 1). Interestingly, using the same primers, we also isolated a new *O*-methyltransferase gene from the cDNA of JX, which was not found in all reported reference genomes of tea plants[19–24]. Considering that JX also accumulates EGCG4″Me[10], we believe that this gene may be responsible for EGCG4″Me biosynthesis. Phylogenetic analysis of these two *O*-methyltransferases with other *O*-methyltransferases found in plants suggests that they are closely related to the flavonol and anthocyanin *O*-methyltransferase (FAOMT) in grape but relatively distant from the CCoAOMT of the tea plant (Fig. 1c)[17]. Therefore, these two *O*-methyltransferases are referred to as CsFAOMT1 and CsFAOMT2.

### EGCG3″Me content is closely associated with *CsFAOMT1*
We next examined the tissue-specific pattern for *CsFAOMT1* expression and EGCG3″Me level. *CsFAOMT1* has the highest transcript level in young leaves, closely followed by flower buds, but much lower in mature leaves, flowers, old leaves, roots, and seeds (Fig. 2a, b). Consistently, high levels of EGCG3″Me were detected in young leaves, followed by flower buds, mature leaves, flowers, and old leaves, while roots and seeds accumulate undetectable amounts of EGCG3″Me, due to the low precursor content of EGCG (Supplementary Table 2) combined with the low transcript level of *CsFAOMT1* (Fig. 2c). One exception is the mature leaves, which showed similar EGCG3″Me content as flower buds, suggesting a gradual decrease in the EGCG3″Me content during leaf aging. Furthermore, the results of quantitative real-time PCR (qRT-PCR) of the 20 individuals from the F$_1$ population (Fig. 2d) and 27 accessions of tea germplasms with different genetic backgrounds (Fig. 2e) confirmed that the transcript levels of *CsFAOMT1* were significantly positively correlated to EGCG3″Me accumulation.

On the basis of the *CsFAOMT1* promoter sequence, we designed primer pairs and cloned the gene. Four genotypes were identified from the individuals of the JX × ZJ F$_1$ population and were named A–D (Supplementary Figure 2). The EGCG3″Me content differed significantly among the four genotypes: genotype A has the highest EGCG3″Me content, about twice that of genotype B and genotype C, whereas they were all significantly higher than that of genotype D, where EGCG3″Me was almost undetectable (Fig. 2f). The *CsFAOMT1* genotype could explain more than 87% of the variation in EGCG3″Me content in spring and autumn. These results suggest that *CsFAOMT1* is a key candidate gene for the regulation of EGCG3″Me biosynthesis.

### CsFAOMT1 has a specific 3″-*O*-methyltransferase activity on EGCG
To identify the enzymatic activity of CsFAOMT1, in vitro methylation assays were conducted with the recombinant protein expressed in *Escherichia coli*, followed by ultra-performance liquid chromatography (UPLC) and liquid chromatograph mass spectrometer (LC−MS) analysis of the products. As shown in Fig. 3a, CsFAOMT1 exhibits significant

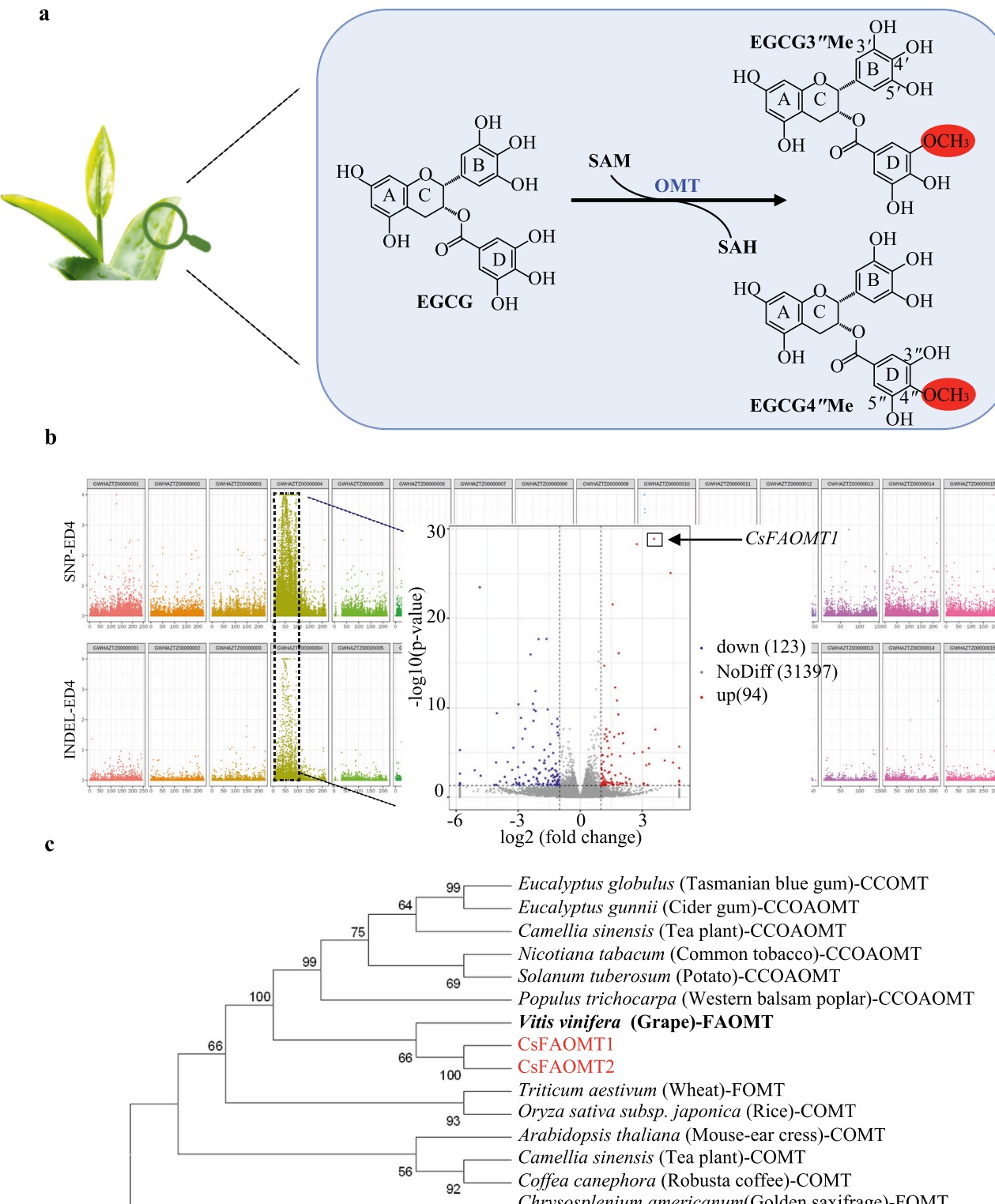

**Fig. 1 | Identification of the key candidate genes involved in the synthesis of *O*-methylated catechins in tea plants. a** Biosynthetic pathway of main *O*-methylated catechins (EGCG3″Me and EGCG4″Me). SAM, *S*-adenosyl-L-methionine. OMT *O*-methyltransferase, SAH *S*-adenosyl-L-homocysteine. **b** Chromosome distribution of ED4 (Euclidean distance (ED) raised to the fourth power) values based on SNPs and InDels and differentially expressed genes (DEGs) in the volcano plot. With the chromosomal location of 'Huangdan' genome used as the abscissa and the ED4 value used as the ordinate, the chromosomal scores of RNA-seq SNP and InDel ED4 values were plotted. The conditions for screening DEGs were |log₂FoldChange| > 1

and $P < 0.05$ by unpaired two-tailed test. **c** Phylogenetic analysis of *O*-methyltransferases. Phylogenetic tree of selected OMTs: *Eucalyptus globulus* CCoAOMT (O81185), *Eucalyptus gunnii* CCoAOMT (O04854), *Nicotiana tabacum* CCoAOMT (O24151), *Solanum tuberosum* CCoAOMT (Q8H9B6), *Camellia sinensis* CCoAOMT (K9USP2), *Populus trichocarpa* CCoAOMT (O65922), *Vitis vinifera* FAOMT (C7AE94), *Triticum aestivum* FOMT (Q84N28), *Oryza sativa subsp. Japonica* COMT (Q6ZD89), *Chrysosplenium americanum* FOMT (Q42653), *Arabidopsis thaliana* COMT (Q9FK25), *Camellia sinensis* COMT (E2FYC3), *Coffea canephora* COMT (Q8LL87).

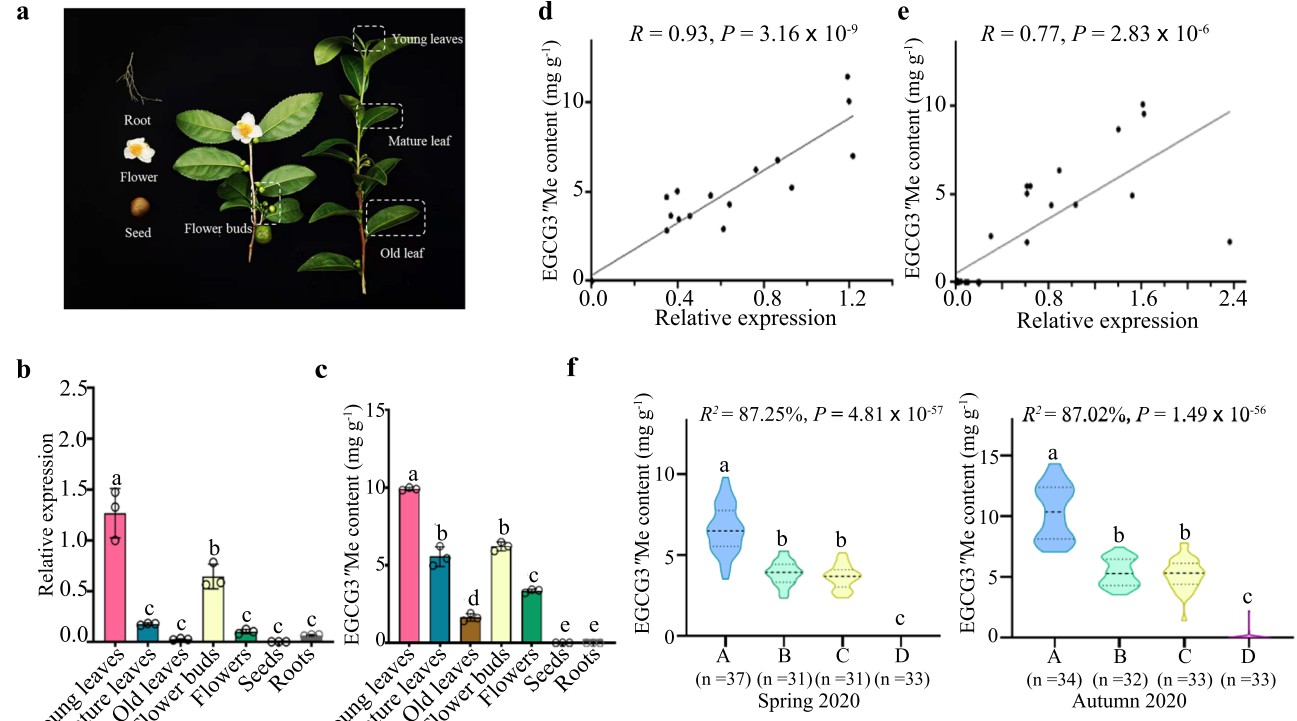

**Fig. 2 | EGCG3″Me content is closely associated with *CsFAOMT1*. a** Different tissues of the tea plant. **b**, **c** Relative transcript levels of *CsFAOMT1* (**b**) and EGCG3″Me contents (**c**) in different tissues. Data represent the mean ± standard deviation of three samples. The corresponding dot plots are overlaid on the figure. The experiment was repeated twice with similar results. Different letters above columns indicate significant differences between tissues using one-way ANOVA at $P < 0.05$ ($P = 2.77 \times 10^{-9}$ and $P = 1.62 \times 10^{-15}$ for (**a**) and (**b**), respectively). **d** Correlation between the transcript levels of *CsFAOMT1* and the contents of EGCG3″Me in 20 individuals of $F_1$ population. **e** Correlation between the transcript levels of *CsFAOMT1* and the content of EGCG3″Me in 27 accessions of tea germplasms. **f** The EGCG3″Me content differences between the four *CsFAOM1* genotypes of the $F_1$ population in two seasons. The EGCG3″Me contents had been identified in our previous study by UPLC[18]. $R^2$, the percentage of the phenotypic variance explained. Different letters indicate significant differences between genotypes using one-way ANOVA at $P < 0.05$.

3″-*O*-methyltransferase activity to convert EGCG to EGCG3″Me in a $Mg^{2+}$-dependent manner. In UPLC, only one product, which has the same retention time as the standard EGCG3″Me, was observed. LC–MS/MS analysis further validated that the product has identical $MS^2$ fragment ions as EGCG3″Me, of which one fragment with an m/z value of 167.09 supports the existence of a methyl group on the D-ring (Supplementary Fig. 3). To confirm that CsFAOMT1 is indeed responsible for the biosynthesis of EGCG3″Me in vivo, EGCG was injected into leaves of *Nicotiana benthamiana* transiently expressing *CsFAOMT1*. As expected, EGCG3″Me was detected from the leaves fed with EGCG through UPLC–MS analysis (Fig. 3b). Overall, these results indicated that CsFAOMT1 is a specific 3″-*O*-methyltransferase of EGCG.

### CsFAOMT2 has 4″-*O*-methyltransferase activity on EGCG

*CsFAOMT2* can only be isolated from the cDNA of tea plants containing EGCG4″Me (Fig. 4a, b). Similar to the correlation between the transcript level of CsFAOMT1 and EGCG3″Me, the transcript level of CsFAOMT2 was also found to be positively related to the amount of EGCG4″Me accumulated in the $F_1$ population (Fig. 4c), the 27 accessions of tea germplasms (Fig. 4d) and different tissues of the tea plant (Fig. 4e, f). In vitro, enzymatic assays with recombinant CsFAOMT2 protein confirmed that CsFAOMT2 has mainly 4″-*O*-methyltransferase activity on EGCG and a lower 3″- *O*-methyltransferase activity (Fig. 4g and Supplementary Fig. 3). Additionally, two trace amounts of dimethylated EGCG products (EGCGdi-Me1 and EGCGdi-Me2) were observed in the reaction solution and further validated by LC–MS/MS analysis (Fig. 4g and Supplementary Fig. 3). Both products exhibited a $MS^2$ fragment with an m/z value of -181, supporting that the dimethylation occurs on the D-ring. However, due to the lack of standard

samples of 3″,4″-dimethyl EGCG and 3″,5″-dimethyl EGCG, we were unable to differentiate between them.

### Substrate scope and kinetic analysis of CsFAOMT1 and CsFAOMT2

Recombinant CsFAOMT1 and CsFAOMT2 enzymes were further tested with a number of other potential phenolic substrates, including catechins ((−)-epicatechin (EC), (−)-epicatechin gallate (ECG)), phenolic acids (gallic acid, methyl gallate and caffeic acid), flavonols (isoquercitrin (quercetin 3-*O*-glucoside) and myricetin) and anthocyanin (cyanidin 3-*O*-galactoside). First, CsFAOMT1 and CsFAOMT2 barely showed activity with EC as substrate, but they readily converted ECG, which has a galloyl group, into its methylated form, as observed on EGCG (Fig. 5, Supplementary Table 3 and Supplementary Fig. 4). For CsFAOMT1, only one methylated form was observed for the other seven substrates except for EC. These results indicated that recombinant CsFAOMT1 specified catalyzed in vitro the 3-*O*-methylation (gallic acid, methyl gallate, and caffeic acid), 3′-*O*-methylation (isoquercitrin, myricetin, and cyanidin 3-*O*-galactoside), and 3″-*O*-methylation (ECG) of substrates with a catechol (di-OH) ring or a pyrogallol (tri-OH) ring. However, these phenolic compounds, except for cyanidin 3-*O*-glucoside, were converted to different monomethyl products by CsFAOMT2 (Fig. 5, Supplementary Table 3 and Supplementary Fig. 4). CsFAOMT2 could methylate ECG at the 3″-and 4″-positions, with ECG4′Me as the dominant product. Tamarixin (tamarixetin 3-*O*-glucoside) and mearnsetin were the main products of isoquercitrin and myricetin, and gallic acid and methyl gallate were almost all converted to 4-*O*-methylgallic acid and methyl 4-*O*-methylgallate, respectively. Moreover, 3′,5′-dimethyl ether (syringetin) and 3′,4′-dimethyl ether were

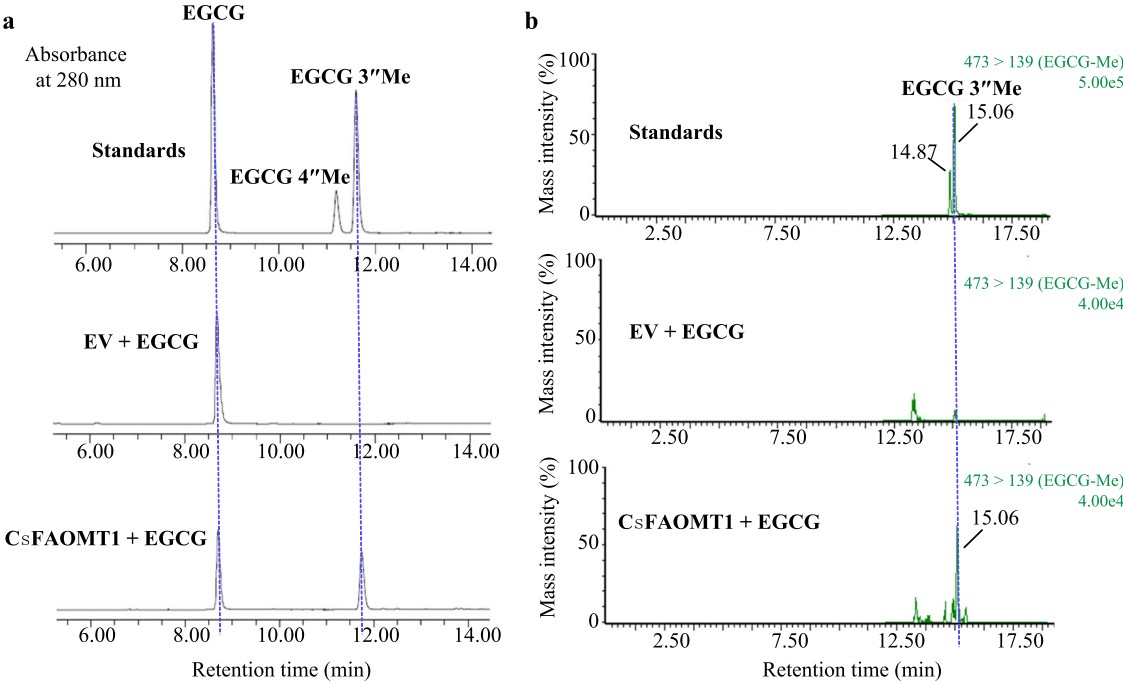

**Fig. 3 | Identification of 3″O-methyltransferase involved in converting EGCG into EGCG3″Me in tea plant. a** In vitro *O*-methyltransferase activity analysis using cell lysates containing the overexpressed *CsFAOMT1*, using EGCG as a substrate and the empty vector (EV) as a negative control. The reaction products were analyzed by UPLC with the absorbance wavelength set at 280 nm. This experiment was replicated twice. **b** In vivo *O*-methyltransferase activity analysis of CsFAOMT1.

EGCG3″Me (473å 139) in the transient transgenic *CsFAOMT1* tobacco leaves was detected by mass spectra, with the transgenic tobacco leaves containing EV as the control. 2.0 mL of 10 mM EGCG was equally injected into the back of three tobacco leaves, and samples were collected after two days and analyzed for *O*-methylated EGCG contents.

obtained with myricetin as substrate (Fig. 5g), which means that CsFAOMT2 could modify myricetin on all the 3′-, 4′-, and 5′-positions of the B ring. Detailed quantifications of reaction products are presented in Supplementary Table 3 and Supplementary Fig. 4.

To verify further the *O*-methylation activity of CsFAOMT1 and CsFAOMT2, kinetic analyses were performed using EGCG as substrate. First, we investigated the optimal reaction conditions and found that CsFAOMT1 has the highest activity at pH 7.4 and 40 °C in the presence of 1.0 mM $Mg^{2+}$ (Supplementary Figure 5a). The best suitable reaction condition for CsFAOMT2 is at pH 7.6 and 45 °C (Supplementary Figure 5b). Under the optimal conditions, the $K_m$ parameters of CsFAOMT1 and CsFAOMT2 for EGCG were measured under saturating *S*-adenosyl-L-methionine (SAM) concentrations (3 and 4 mM, respectively) using the highly purified recombinant proteins expressed in *E. coli* (Supplementary Fig. 6). The $K_m$ values, 95.79 μM for CsFAOMT1 and 20.88 μM for CsFAOMT2, indicated that CsFAOMT2 has a stronger affinity to EGCG than CsFAOMT1 (Fig. 5 and Supplementary Table 4). In addition, the $k_{cat}/K_m$ value of CsFAOMT2 (6762.45 $s^{-1} M^{-1}$) was higher than that of CsFAOMT1 (3830.57 $s^{-1} M^{-1}$), suggesting that CsFAOMT2 has higher catalytic efficiency.

The substrate preferences of CsFAOMT1 and CsFAOMT2 were further characterized by determining the kinetic constants for ECG, gallic acid, methyl gallate, caffeic acid, isoquercitrin, myricetin, and cyanidin 3-*O*-galactoside (Fig. 6). Given the obvious inhibitory effect of higher SAM concentrations, the kinetic analysis of CsFAOMT1 for caffeic acid, isoquercitrin, and cyanidin 3-*O*-galactoside was performed at a lower saturating SAM concentration (1 mM) (Supplementary Table 5), and CsFAOMT2 for ECG, gallic acid, caffeic acid, isoquercitrin, and cyanidin 3-*O*-galactoside were measured in the presence of 2 mM SAM (Supplementary Table 6). The results showed that CsFAOMT1 displayed higher affinity and substantially higher efficiency toward ECG compared with EGCG, which were reflected by lower $K_m$ and higher $k_{cat}/K_m$ values (Fig. 6a, b and Supplementary

Table 4). However, the content of ECG3″Me is positively correlated with but significantly lower than that of EGCG3″Me in tea plant[10,25], which may be because of the lower content of ECG. CsFAOMT1 displayed low $K_m$ (<4 μM) and high $k_{cat}/K_m$ ($k_{cat}/K_m$ > 39,000 $s^{-1} M^{-1}$) values for methyl gallate and myricetin (Fig. 6d, g), indicating strong specific activities toward these two substrates. For isoquercitrin, a considerably high $K_m$ value (>2500 μM) was observed (Fig. 6f), resulting in poor catalytic efficiency ($k_{cat}/K_m$ = 225.72 $s^{-1} M^{-1}$). CsFAOMT2 showed a narrower range of $K_m$ (10.06–144.50 μM) and a wider range of $k_{cat}$ (28.54–623.43 × $10^{-3} S^{-1}$) values for these seven substrates compared with CsFAOMT1 (Fig. 6 and Supplementary Table 4). For CsFAOMT2, ECG exhibited similar kinetic parameters to that of EGCG (Fig. 6a, b). Low $k_{cat}$ (28.54 ± 3.06 × $10^{-3} S^{-1}$) and $k_{cat}/K_m$ values (361.95 $s^{-1} M^{-1}$) were observed using cyanidin 3-*O*-galactoside as substrate (Fig. 6h), which showed a weaker catalytic efficiency.

## Crystal structures of CsFAOMT1 and CsFAOMT2 reveal key residues for substrate recognition

To explain the specificity of CsFAOMT1 and CsFAOMT2, we crystallized both in the presence of EGCG and *S*-adenosyl-L-homocysteine (SAH), a by-product of the cofactor SAM and determined the structures at a resolution of 2.0 Å for CsFAOMT1 and 2.0 Å for CsFAOMT2 (Supplementary Fig. 7 and Supplementary Table 7). Both structures exhibited the typical Rossmann fold and a dimeric arrangement similar to that of other cation-dependent OMTs[26]. The electron densities for SAH and $Mg^{2+}$ are unambiguously assigned, but EGCG was not found in either structure. The structural overlap of the monomers of CsFAOMT1 and CsFAOMT2 revealed a root mean square deviation of 0.36 Å over the Cα atom of 190 aligned residues, with the most striking difference occurring in an α-helix (α9) near the substrate binding pocket (Fig. 7a). In CsFAOMT1, this α-helix is bent toward the pocket, creating a deep and narrow cleft for substrate binding, whereas in CsFAOMT2 the corresponding α-helix is bent in the opposite direction, resulting in a

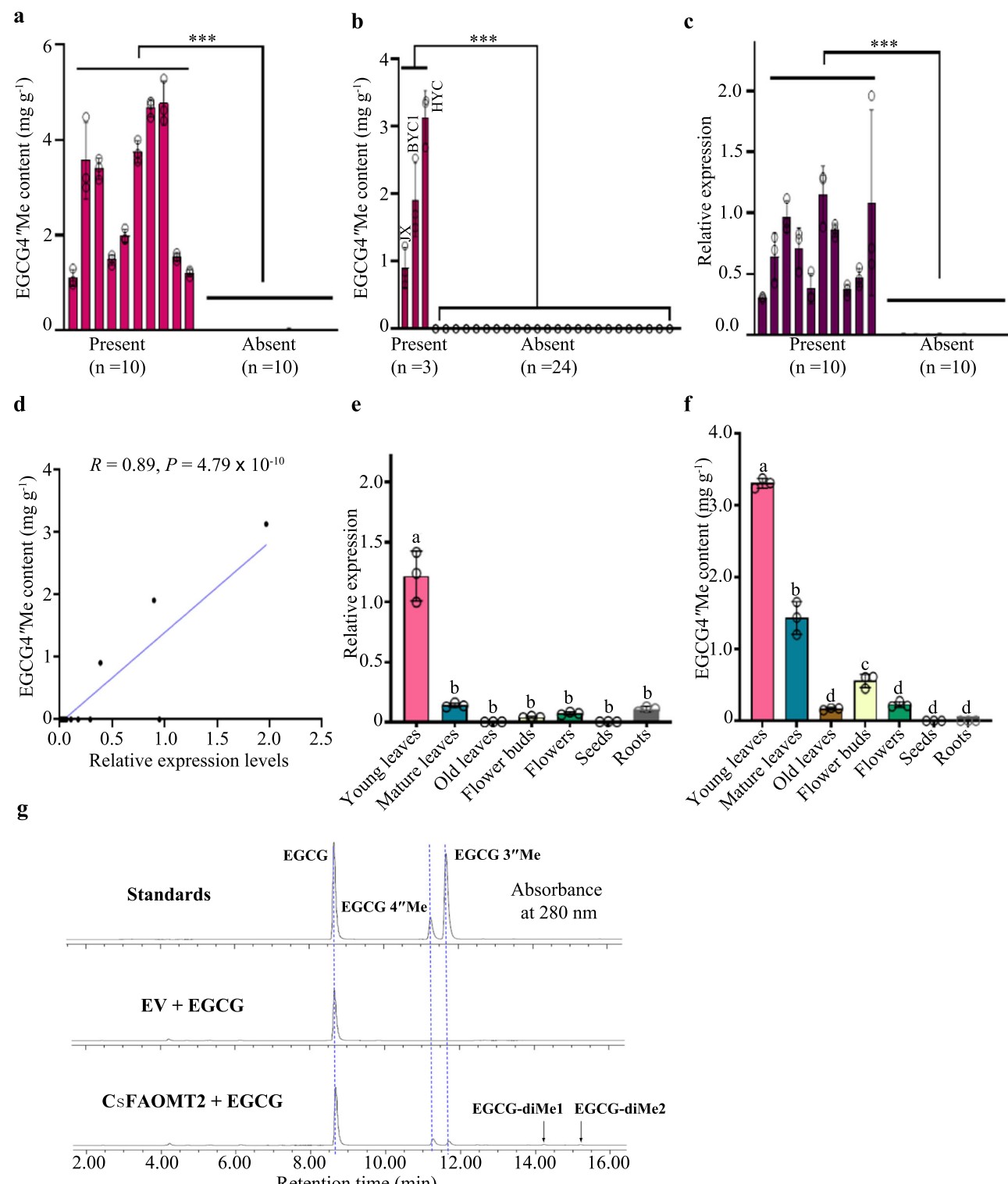

relatively open and shallow pocket. Comparison with other *O*-methyltransferases showed that CsFAOMT1 and CsFAOMT2 share high structure similarity with three plant-derived *O*-methyltransferases: a Phenylpropanoid and Flavonoid *O*-methyltransferase (McPFOMT) from *Mesembryanthemum crystallinum* (PDB code 3C3Y)[26] and two CCoAOMT from *Medicago sativa* (PDB code 1SUI)[27] and *Sorghum bicolor* (PDB code 5KVA)[28]. All these structures contain an *N*-terminal arm that precedes the Rossmann fold, which has been shown to be critical for substrate binding. Moreover, the bent α-helix is present in all these plant *O*-methyltransferases and is longer than the

corresponding helix in other cation-dependent *O*-methyltransferases, such as the Catechol *O*-methyltransferases (COMT) and CCoAOMT-like, from bacteria and animals[29–31] (Supplementary Fig. 8). Sequence alignment also supports a closer relationship between CsFAOMTs and the plant-derived *O*-methyltransferases, with sequence identities of >53%, much higher than the <35% between CsFAOMTs and other OMTs from bacteria and animals (Supplementary Fig. 9).

Given that CsFAOMT1 and CsFAOMT2 share more than 90% sequence identity (Supplementary Fig. 1), their different positional preferences could be due to minor variable residues subtly affecting

**Fig. 4 | Identification of 4″ O-methyltransferase involved in converting EGCG into EGCG4″Me in tea plant. a** Differences of EGCG4″Me content between ten individuals containing *CsFAOMT2* (Present) and ten individuals lacking *CsFAOMT2* (Absent) in JX × ZJ F₁ population. ***denotes a significant difference between the two groups using a two-tailed Student's *t*-test at $P < 0.001$ ($P = 1.06 \times 10^{-5}$). **b** Differences of EGCG4″Me content between three accessions containing *CsFAOMT2* (Present) and 24 accessions lacking *CsFAOMT2* (Absent) in the 27 tea germplasms with different genetic backgrounds. ***denotes a significant difference between the two groups using a two-tailed Student's *t*-test at $P < 0.001$ ($P = 1.83 \times 10^{-10}$). **c** The transcript level differences of *CsFAOMT2* between ten individuals containing *CsFAOMT2* (Present) and ten individuals lacking *CsFAOMT2* (Absent) in the F₁ population. ***denotes a significant difference between the two groups using a two-tailed Student's *t*-test at $P < 0.001$ ($P = 2.87 \times 10^{-15}$). **d** The correlation analysis between the transcript levels of *CsFAOMT2* and the contents of EGCG4″Me in the 27 accessions of tea germplasms. **e, f** Relative transcript levels of *CsFAOMT2* (**e**) and EGCG4″Me contents (**f**) in different tissues. Data represent the mean ± standard deviation of three samples. The corresponding dot plots are overlaid on the figure. The experiment was repeated twice. Different letters indicate significant differences between tissues using one-way ANOVA at $P < 0.05$ ($P = 1.96 \times 10^{-10}$ for (**e**) and $P = 2.87 \times 10^{-15}$ for (**f**), respectively). **g** In vitro *O*-methyltransferase activity analysis of CsFAOMT2 in cell lysates, using EGCG as a substrate and the empty vector (EV) as a negative control. The reaction products were analyzed by UPLC, with the absorbance wavelength set at 280 nm. Two dimethylated EGCGs (EGCGdi-Me1 and EGCGdi-Me2) were labeled.

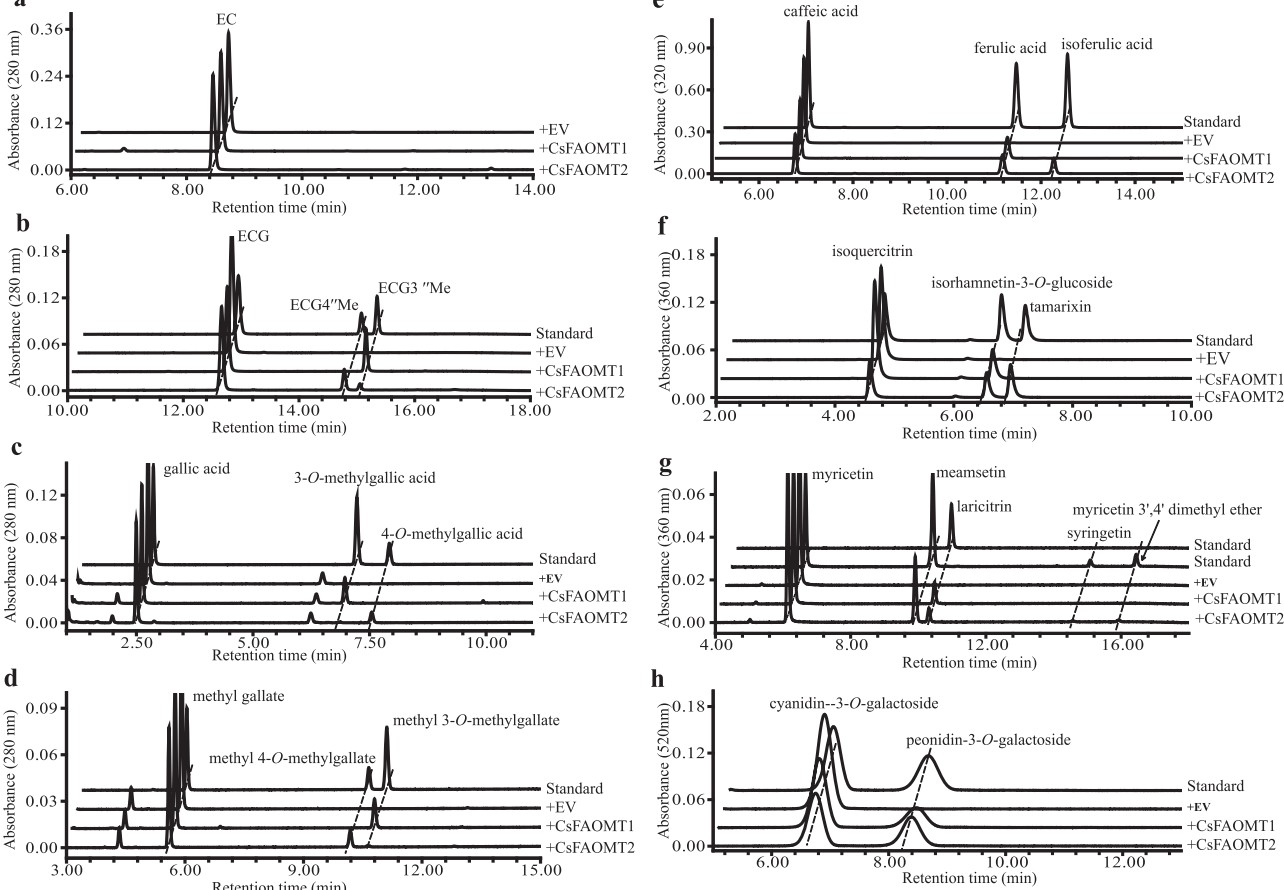

**Fig. 5 | UPLC chromatograms of the methylation of catechins, phenolic acids, flavonols, and anthocyanin by CsFAOMT1 and CsFAOMT2 in vitro. a** (−)-epicatechin (EC, 0.4 mM); **b** (−)-epicatechin gallate (ECG, 0.4 mM); **c** gallic acid (0.4 mM); **d** methyl gallate (0.4 mM); **e** caffeic acid (0.2 mM); **f** isoquercitrin (0.2 mM); **g** myricetin (0.4 mM); **h** cyanidin 3-*O*-galactoside (0.2 mM). EV empty vector. MS/MS data of products and their authentic compounds are indicated in Supplementary Fig. 5 and Supplementary Table 3.

the local chemical environment or conformation. We paid particular attention to the residues surrounding the Mg²⁺ because they may play a crucial role in coordinating the D-ring of EGCG and aligning the target hydroxyl group with the Mg²⁺. This step is critical for the subsequent attack of the ionized hydroxyl group on the methyl group of SAM. Two sequence variable residues were identified subsequently: CsFAOMT1-L50 (CsFAOMT2-M50) and CsFAOMT1-Y184 (CsFAOMT2-R184) (Supplementary Fig. 7c). In vitro enzymatic assays were performed with CsFAOMT1 and CsFAOMT2 mutants exchanging these two residues, using wild-type CsFAOMT1 and CsFAOMT2 as controls (Fig. 7b). Interestingly, CsFAOMT1-L50M and CsFAOMT1-Y184R led to totally different results in 3″-*O*-methylation activity: a moderate increase by CsFAOMT1-L50M but a significant decrease by CsFAOMT1-

Y184R; the activity of CsFAOMT1-Y184R can be increased about twice (172%) when CsFAOMT1-L50M is combined, but still much lower than that of the wild-type CsFAOMT1. However, CsFAOMT1-Y184R acquired the ability to methylate the 4″-position of EGCG, with approximately 25% of the activity of CsFAOMT2. The 4″-*O*-methylation activity of CsFAOMT1-Y184R could be further increased when CsFAOMT1-L50M was incorporated, although CsFAOMT1-L50M alone showed no 4″-*O*-methylation activity at all. Consistently, CsFAOMT2-R184Y led to significant increase of 3″-methylation activity and moderate decrease of the ability to methylate the 4″-position of EGCG, and the 3″- and 4″-*O*-methylation activities of CsFAOMT2-R184Y were decreased by 58% and 54% when M50L mutation was combined in CsFAOMT2, respectively (Fig. 7b).

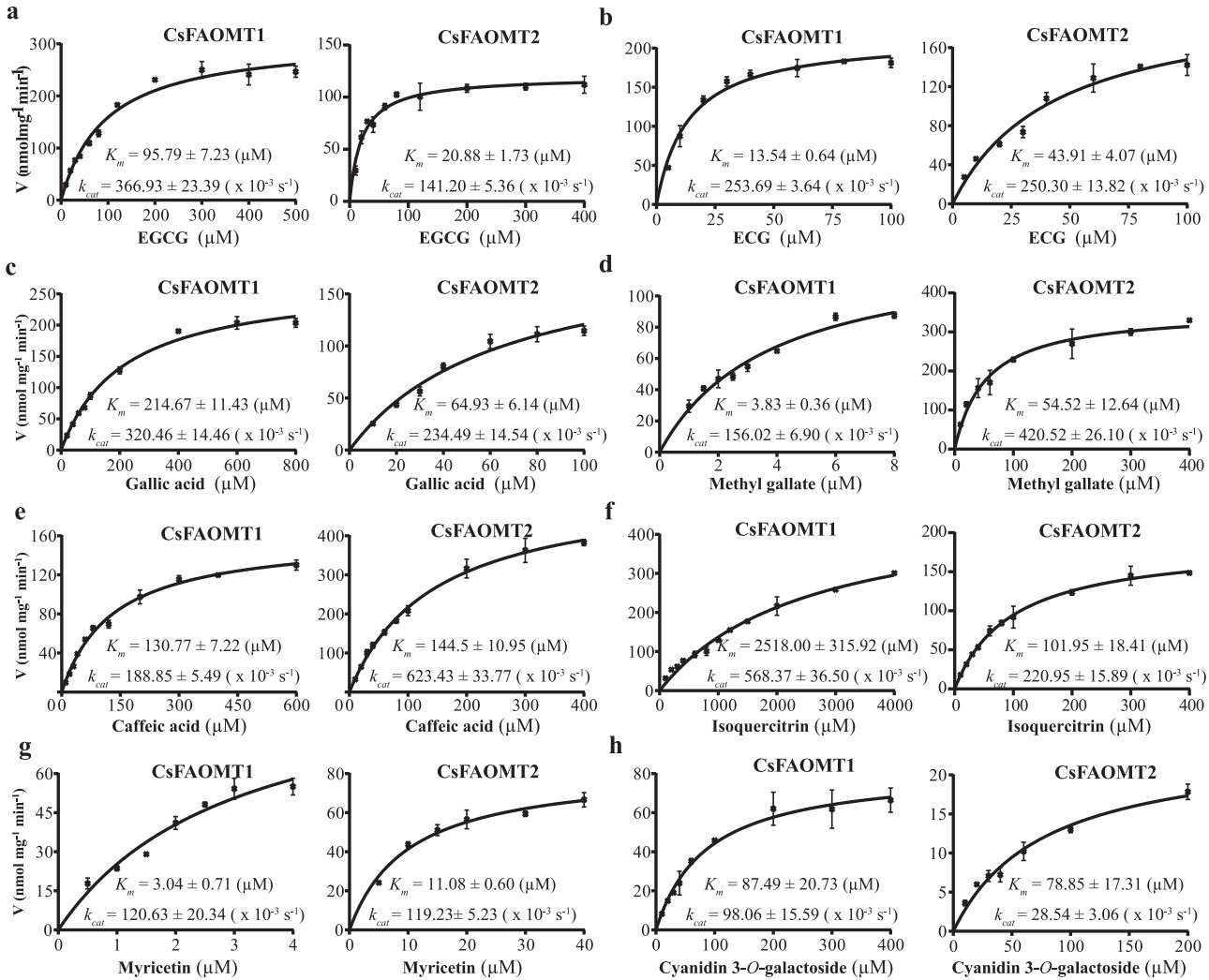

**Fig. 6 | Steady-state kinetic analysis of CsFAOMT1 and CsFAOMT2.** EGCG (**a**), ECG (**b**), gallic acid (**c**), methyl gallate (**d**), caffeic acid (**e**), isoquercitrin (**f**), myricetin (**g**), and cyanidin 3-O-galactoside (**h**) were used as substrates, respectively. Figure data are means ± SD of three replicates. The black line represents the nonlinear least-squares fit of the initial velocities versus substrate concentration to the hyperbolic Michaelis–Menten equation. Kinetic parameters of CsFAOMT1 and CsFAOMT2 were determined at saturating concentrations.

To clarify the exact roles of these residues in substrate binding, we replaced SAH with SAM and then docked EGCG in the structure of CsFAOMT1. EGCG is embedded in a narrow and deep cleft formed by two α-helices, α3, and the bent α-helix α9, with both the A-ring and B-ring anchored through hydrogen bonds (Fig. 7a and Supplementary Figure 10). Briefly, Y46 from α3 interacts with the 7-OH on the A-ring. For the B-ring, K156 and two residues from α9, R197 and S201, recognize 3'-, 4'- and 5'-OH, respectively. We also docked EGCG into the structures of CsFAOMT1-L50M, and CsFAOMT1-Y184R generated through homology modeling. Notably, both CsFAOMT1-Y184 and CsFAOMT1-Y184R could form a hydrogen bond with the carbonyl group on the galloyl moiety, thereby positioning the D-ring into the active site, which may account for the specificity of CsFAOMTs to the D-ring of EGCG. The difference is that CsFAOMT1-Y184 helps to locate the 3″-hydroxyl on the D-ring close to the methyl group on SAM (Fig. 7c); substituting the tyrosine with an arginine that contains a longer side chain pushes the D-ring into a deeper pocket, resulting in the replacement of 3″-hydroxyl group by 4″-hydroxyl group close to the methyl group on SAM (Fig. 7d). The CsFAOMT1-L50M mutation does not trigger the movement of EGCG (Fig. 7e). Instead, it possibly contributes to the stabilization of the galloyl group of EGCG in the binding pocket, thus increasing the 3″-methylation activity of CsFAOMT1 and CsFAOMT1-Y184R. Furthermore, we found a hydrophilic residue E49 in CsFAOMT1 that is in the proximity of the 5″-OH of EGCG and might be involved in the recognition of it through water-mediated hydrogen bonds (Supplementary Figs. 10 and 11), explaining why only EGCG3″Me was generated by CsFAOMT1. However, in CsFAOMT2, this residue is mutated to a hydrophobic valine (V49), which makes it possible to accommodate 5″-OMe and leads to the production of dimethyl EGCG.

The position specificities of CsFAOMT1 and CsFAOMT2 are reminiscent of that of McPFOMT[26] and SynOMT[31], respectively. When using 3,4,5-trihydroxycinnamic acid as a substrate, McPFOMT only performs O-methylation on the ortho-OH (3-OH), while SynOMT could methylate both the ortho- and para-OH (4-OH) groups. Comparison of their active sites revealed similar roles of CsFAOMT1-Y184 equivalences: like the replacement of Y184 in CsFAOMT1 with R184 in CsFAOMT2, G185 in PFOMT is substituted by H174 in SynOMT to push the 4-OH group closer to the $Mg^{2+}$ (Supplementary Fig. 11).

## Discussion

O-methylated catechins are attracting widespread interest because they are more stable, bioavailable, and potent than their parent compounds[9]. The natural content of O-methylated catechins in tea

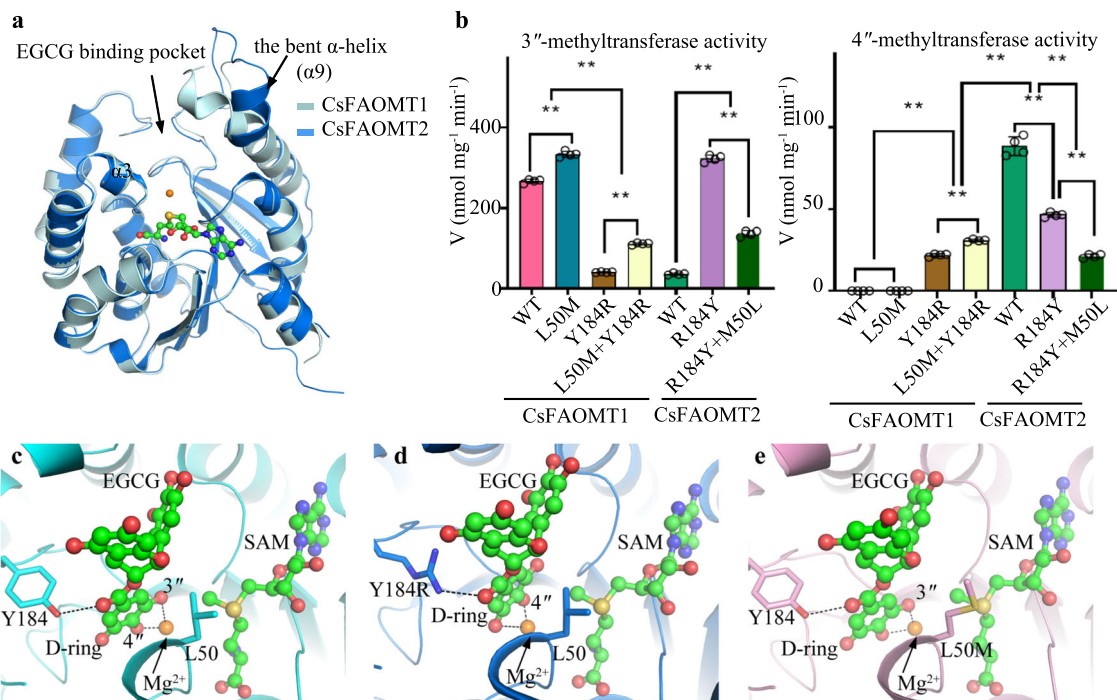

**Fig. 7 | Crystal structure of CsFAOMT1 and CsFAOMT2. a** Structural overlap of CsFAOMT1 and CsFAOMT2. CsFAOMT1 and CsFAOMT2 are colored pale cyan and blue, respectively. The Mg$^{2+}$ is shown as a brown sphere. The SAH is shown as a ball-and-stick presentation. **b** In vitro methylation assay of the mutants of CsFAOMT1 and CsFAOMT2 using EGCG as substrate. Data represent mean ± SD ($n = 4$). The corresponding dot plots are overlaid on the figure. The experiment was repeated twice. Differences were assessed statistically by a two-tailed Student's $t$-test. **$P < 0.01$. **c**–**e** Molecular docking with EGCG and SAM into the substrate binding pocket of CsFAOMT1 (**c**), CsFAOMT1-Y184R (**d**), and CsFAOMT1-L50M (**e**). The D-ring of EGCG, 3″- and 4″- positions are labeled. The side chains are shown as sticks.

plants is relatively low, making it difficult to exploit their health-promoting effects. Large-scale cultivation of tea plants with high levels of $O$-methylated catechins to pool elite genes in tea germplasms could be an effective measure, but more information about the enzymes involved in the biosynthesis of $O$-methylated catechins is needed. In this study, we identified CsFAOMT1, which has a methylation activity at the 3″-position, and CsFAOMT2, which is a dominant methyltransferase at the 4″-position of galloylated flavan-3-ols (EGCG and ECG). Moreover, we found that the levels of EGCG3″Me and EGCG4″Me in tea plants are positively correlated with the transcript levels of CsFAOMT1 and CsFAOMT2, respectively. By aggregating elite alleles of *CsFAOMT1* and *CsFAOMT2* with high transcript levels, it is possible to innovate elite tea germplasms with higher levels of $O$-methylated catechins in the future. The crystal structures of CsFAOMT1 and CsFAOMT2, in combination with in vitro enzymatic assays, have revealed key residues that distinguish the 3″- and 4″-positions of EGCG during catalysis. After the genetic transformation technology is mature in the tea plant, this information will enable rational modification of $O$-methyltransferases to breed new tea cultivars with high $O$-methylated EGCG contents. Therefore, this study provides important clues for the future breeding of tea plants high in $O$-methylated catechins.

Substrate binding and specificity are of great interest in plant $O$-methyltransferases because they are involved in the biosynthesis of numerous compounds with diverse structures and biological significance. The exchange of a single residue in CsFAOMT1 (Y184) with the corresponding residue in CsFAOMT2 (R184) appears to be sufficient to confer apparent 4″-methyltransferase activity on CsFAOMT1, although the activity is much lower than that of CsFAOMT2. Notably, the side chains of arginine are known to be more flexible than those of tyrosine, meaning that EGCG may bind in a more dynamic manner at the active site of CsFAOMT2 and CsFAOMT1-Y184R mutant, thus conferring them 3″- and 4″-$O$-methylation activity. Other residues surrounding the substrate binding pocket may also contribute to

positional specificity, particularly those on the bent α-helix. The specificity of $O$-methyltransferases results from the interplay of multiple factors and is not solely due to the enzyme. These also include the molecular structure and properties of the substrate, as well as the reaction conditions, but may not be limited to these features. Future work to understand how this is achieved in CsFAOMT1 will be critical to exploit the secondary metabolism of EGCG.

## Methods

### Chemicals

$S$-adenosyl-$_L$-methionine (SAM) was purchased from Shenggong (Shanghai, China). Gallic acid, EC, (−)-epigallocatechin, ECG, and EGCG were purchased from Sigma-Aldrich (St. Louis, MO, USA). ECG3″Me, ECG4″Me, EGCG3″Me, and EGCG4″Me, were purchased from Nagara Science Co., Ltd. (Gifu, Japan). Myricetin and tamarixetin were obtained from BioBioPha Co., Ltd (Kunming, China), and caffeic acid, ferulic acid, isoferulic acid, 3-$O$-Methylgallic acid, 4-$O$-Methylgallic acid, methyl gallate, methyl 4-$O$-methylgallate, cyanidin 3-$O$ galacto-side chloride, peonidin 3-$O$-galactoside chloride, isoquercitrin, iso-rhamnetin 3-$O$-glucoside and syringetin were purchased from Shanghai Yuanye Bio-Technology Co., Ltd. (Shanghai, China). Mearn-setin and laricitrin were from ChemFaces (Wuhan, China). Methyl 3-$O$-methylgallate was from MedChemExpress (Monmouth Junction, NJ, USA). Myricetin 3′, 4′-dimethyl ether was provided by Prof. Ai-Xia Chen of Shandong University, China.

### Plant materials

A hybrid population consisting of more than 150 F$_1$ individuals was used in this study. This population was constructed via artificial hybridization using parents 'Jinxuan' (*Camellia sinensis* var. *sinensis*, JX) and 'Zijuan' (*C. sinensis* var. *assamica*, ZJ). Young shoots of "one bud and one leaf" were collected from each individual in the hybrid population and their parents. In addition, 27 accessions of tea

germplasms with wide genetic variation, which contain the *O*-methylated catechins[10], were selected for this study. Samples of different tissues (buds, leaves, flowers, roots, and seeds) were collected from the male parent JX and stored at −80 °C immediately. The materials were used for the extraction of genomic DNA and RNA and the analysis of catechins. The tobacco used for transient transformation was *N. benthamiana*.

## Analysis of catechins in the tea plant

The samples, including tea leaves, buds, leaves, flowers, roots, and seeds, were lyophilized and ground with a RETSCH MM400 grinder at a speed of 25–30 r min⁻¹ for 2 min. About 0.1 g of each sample tissue was soaked in 10 mL of 70% (v/v) methanol (Thermo Fisher, USA) to sonicate for 30 min and then let stand at 4 °C for 2 h. The supernatant was filtered by a 0.22 μm nylon filter before LC-MS/MS analysis on a UPLC system (H-Class, Waters, MA, USA) coupled to a triple quadrupole mass spectrometer (TQS-Micro, Waters, MA, USA). Chromatographic separation of the tea metabolites was performed on an ACQUITY UPLC HSS T3 C18 column (1.8 μm, 2.1 mm × 100 mm, Waters, USA) using phase (A) of 0.1% (v/v) formic acid (TIC, Tokyo, Japan) and phase (B) of acetonitrile (Thermo Fisher, USA). The chromatographic elution was set as follows: 0 min, 2% B; 5 min, 2% B; 10 min, 10% B; 15 min, 20% B; 17 min, 25% B; 17.1 min, 100% B; 18.6 min, 100% B; 18.7 min, 2% B; 20 min, 2% B. The flow rate was set at 0.28 mL/min. The injection volume was 1 μL. Electrospray ionization (ESI) was performed in the positive ionization mode with the following parameters: the capillary voltage was set as 3 kV; the temperature of the ion source was set at 600 °C; the flow rates of the cone gas and the desolvation gas were set at 50 and 600 L/h, respectively. The optimized multiple reaction monitoring transitions and collision energy voltages are shown in Supplementary Table 8. The metabolites were absolutely quantified using the calibration curves of the corresponding standard solutions. The contents of *O*-methylated products in Hongyacha (HYC), a tea genetic resource recently found in Fujian of China[32], also contain GCG3″Me and GCG4″Me, respectively.

## RNA-seq analysis

Total RNAs were extracted following the manufacturer's protocol of the EASY-spin Plus Complex Plant RNA kit (Aidlab Biotechnologies Company, Beijing, China) and then measured using NanoDrop (Thermo, Waltham, MA). Transcriptome sequencing was completed by Shanghai Personal Biotechnology Co., Ltd. An Illumina NovaSeq platform (Illumina, San Diego, CA, USA) was used to conduct RNA-seq experiments. A total of 24 individuals with high EGCG3″Me content (>5.0 mg g⁻¹) and 24 individuals with low EGCG3″Me content (<0.1 mg g⁻¹) in the JX × ZJ F₁ population were selected to form the two extreme groups (group H/L) and for BSR-Seq analysis. The same equivalent of total RNAs of each of the eight individuals from the 24 individuals in group H/L were pooled and treated as a biological replicate (pools H1, H2, and H3, pools L1, L2, and L3), respectively. RNA-seq was conducted for each RNA pool. With 'Huangdan' taken as the reference genome, preliminarily candidate genes were determined in accordance with the gene expression differences between the two groups and combined with the target trait mapping results. Genes with different expression levels were analyzed by DESeq (1.30.0)[33]. The detection of SNPs was mainly performed via the GATK software toolkit[34]. Reads from bulk RNA-seq were used to calculate Euclidean distance (ED) for bulk segregant analysis[35]. Moreover, the 27 accessions of tea germplasms with diverse genetic backgrounds were selected as research materials to perform RNA-seq and screen the key candidate genes. The correlation between gene expression and EGCG3″Me content was calculated based on the Pearson correlation.

## Cloning of the CsFAOMTs gene sequence

On the basis of the instructions of the FastKing cDNA First-Strand synthesis kit (TIANGEN), 200 ng of RNA was used to synthesize first-strand cDNA. The candidate gene was screened, and two pairs of primers were designed on the basis of the sequences of 5′ untranslated coding regions (UTR) and 3′ UTR (Supplementary Table 1). Polymerase chain reaction (PCR) was performed using KOD-Pus-Neo (TOYOBO) in 50 μL of the reaction system. PCR products were detected on a 0.8% agarose gel. Subsequently, the target gene was ligated into pUCm-T by homologous recombination and transferred into *E. coli* (DH5α) for sequence verification. Primer pairs for genotype identification (Supplementary Table 1) were designed with reference to the *CsFAOMT1* promoter sequence, and each individual in the population was genotyped by direct sequencing of PCR products and detecting SNPs that appeared in the promoter.

## Phylogenetic analysis

The neighbor-joining tree was generated using MEGA X software (http://www.megasoftware.net). The amino acid sequences of CsFAOMT1 and CsFAOMT2 homologs were retrieved from the National Center for Biotechnology Information and Universal Protein database (NCBI, https://www.ncbi.nlm.nih.gov/, UniProt, https://www.uniprot.org/) and aligned using CLUSTALW (https://www.genome.jp/tools-bin/clustalw).

## Transcriptional level analysis of *CsFAOMTs* in the tea plant

qRT-PCR was implemented to detect the transcript levels of *CsFAOMTs* in different tea plants and in different tissues. Light-Cycler® 480 System with LightCycler® 480 SYBR Green I Masterkit (Roche Co., Ltd., Mannheim, Germany) was used to detect the transcript levels. The qRT-PCR amplification procedure was 98 °C for 2 min, followed by 45 cycles of 94 °C for 10 s, 58 °C for 15 s, and 72 °C for 12 s. Relative gene expression was calculated using the 2⁻ᐞᐞᶜᵀ method using *Cs18S* as the internal reference gene. All primers are listed in Supplementary Table 1.

## Transient expression in tobacco

*CsFAOMT1* and *CsFAOMT2* were cloned into the 35S-GFP vector and then transformed into Agrobacterium strain GV3101. Approximately 100 μL of *Agrobacterium tumefaciens* were transferred into 10 mL of liquid LB medium to expand to an OD₆₀₀ value of 1.0–1.2. The pellets were collected by centrifugation and then injected into tobacco leaves. After cultivation of the transient transgenic tobacco for two days, 2.0 mL of 10 mM EGCG was injected into the back of the three tobacco leaves, and samples were collected after two days and analyzed for *O*-methylated EGCG contents.

## Protein purification

For in vitro enzymatic activities and kinetic analysis: Genes encoding CsFAOMT1 and CsFAOMT2 were cloned into pMAL-c5x using ClonExpress® II One Step Cloning Kit (Vazyme Biotech Co., Ltd., Nanjing). The plasmids were transferred into *BL21* (DE3) *pLys* cells (TransGen Biotechnologies Co., Beijing, China) to express target proteins containing an N-terminal MBP tag. The cells were incubated at 37 °C for 3–4 h until the cell density reached an OD₆₀₀ of 0.6, and the protein expression was induced by adding 0.5 mM isopropyl β-ᴅ-1-thiogalactopyranoside (IPTG) and incubated at 37 °C for another 4–6 h. Cells were harvested, resuspended in buffer 1 (20 mM Tris-HCl, pH 7.4, 1.0 mM EDTA, 1.0 mM DTT, 200 mM NaCl, 10% (v/v) glycerin), and lysed by sonication (225 W, 10 min). The recombinant proteins were purified using MBP-tagged Amylose resin (New England Biolabs, NEB) and eluted by buffer 2 (20 mM Tris-HCl, pH 7.4, 1.0 mM EDTA, 1.0 mM DTT, 200 mM NaCl, 10% (v/v) glycerin and 10 mM maltose).

For crystallization and structure determination: Genes encoding CsFAOMT1 and CsFAOMT2 were inserted into a modified *pRSF-Duet1* vector, respectively, in which a His$_6$-SUMO tag is inserted ahead of the multicloning sites. The recombinant proteins were overexpressed in *BL21* (DE3) cells at 16 °C overnight by adding IPTG to a final concentration of 0.4 mM when the cell density reached an OD$_{600}$ of 1.0. The cells were harvested, resuspended, and lysed in buffer A (50 mM Tris, pH 8.0, 1 M NaCl, 25 mM imidazole and 5 mM β–Me). The His$_6$-SUMO tagged proteins were first purified through a nickel column, and the tag was cleaved by ULP1. The target proteins were further fractioned by anion exchange chromatography (Hitrap Q, GE Healthcare) and size-exclusive chromatography (Superdex 75 16/600, GE Healthcare). The proteins were concentrated to about 20 mg mL$^{-1}$ in buffer B (25 mM Tris, pH 8.0, 100 mM NaCl and 5 mM DTT) and stored at −80 °C for future use.

## In vitro assay for CsFAOMTs

The in vitro enzymatic activity assays were measured at pH 7.4 and 40 °C for 30 min. Briefly, a 200 µL reaction mixture contained 100 mM Tris-HCl (pH 7.4), 0.5 mM SAM, 4 mM DTT, 0.2 mM or 0.4 mM substrates, 1 mM MgCl$_2$, and 50 µL crude recombinant protein. The reaction was stopped by adding equal volumes of 200 mM HCl (containing 10% methanol and 10% dimethyl sulfoxide). The pMAL-c5x empty vector was used as a negative control. The reaction mixture was centrifuged to remove the precipitate (2 min, 12,000 r min$^{-1}$), and then 10 µL supernatant was analyzed using the same UPLC-UV detection method as our previous study[18]. By contrast, the chromatographic elution of isoquercitrin and myricetin was set as follows: 0 min, 15% B; 15 min, 70% B; 15.1 min, 100% B; 16.5 min, 100% B; 16.6 min, 15% B; 18 min, 15% B. The flow rate was set at 0.4 mL/min. Detection was by UV wavelength at 280 nm (EC, ECG, EGCG, gallic acid, and methyl gallate), 320 nm (caffeic acid), 360 nm (isoquercitrin and myricetin), and 520 nm (cyanidin 3-*O*-galactoside).

The optimal reaction conditions for recombinant proteins were determined after evaluating the effects of different reaction temperatures, time, pH, and ion and SAM concentrations using EGCG as substrate. The catalytic efficiencies of the enzymes were calculated on the basis of the peak area of the product. The in vitro enzymatic activity assays of the wild types and mutants of CsFAOMT1 and CsFAOMT2 were finally measured at pH 7.4 and 40 °C for 10 min. Briefly, a 200 µL reaction mixture contained 100 mM Tris-HCl (pH 7.4), 3 mM SAM, 4 mM DTT, 0.4 mM EGCG, 1 mM MgCl$_2$, and 5 µg purified recombinant protein.

Identification of reaction products produced by recombinant CsFAOMT1 and CsFAOMT2 by LC−MS/MS used the same instrument and chromatographic elution condition as that for the analysis of catechins in the tea plant. By contrast, the chromatographic elution condition of isoquercitrin and myricetin was the same as that used for UPLC-UV. The injection volume was 10 µL. ESI was performed in the positive or negative ionization mode with the following parameters: the capillary voltage was set as 3.5 kV; the temperature of the ion source was set at 500 °C; the flow rates of the cone gas and desolvation gas were set at 50 and 1000 L/h, respectively. The collision energy voltage is shown in Supplementary Table 9.

## Kinetic analysis

The kinetic parameters of CsFAOMTs were measured under optimal conditions, and the reactions were performed in a final volume of a 200 µL system containing 1 µg of purified enzymes (except that 0.1, 0.2, and 3 µg of CsFAOMT1 were used when testing myricetin, methyl gallate, and cyanidin 3-*O*-galactoside, respectively; 0.5 and 3 µg of CsFAOMT2 were used when testing myricetin and cyanidin 3-*O*-galactoside, respectively) and varying concentrations of substrate for 10 min (except for cyanidin 3-*O*-galactoside with 30 min). All reaction products were analyzed using the UPLC-UV system. Enzyme kinetic parameters ($K_m$, $V_{max}$, $k_{cat}$, $k_{cat}/K_m$) were analyzed by the Michaelis−Menten equation by nonlinear regression analysis using GraphPad Prism 8.

## Crystallization and structure determination

The crystallization of CsFAOMT1 and CsFAOMT2 was conducted using the hanging-drop diffusion method at 4 °C and 16 °C, respectively. The proteins were mixed with 1 mM SAH, 10 mM EGCG, and 5 mM MgCl$_2$ and then incubated on ice for 10 min to form complex. Crystals of CsFAOMT1 were obtained in a buffer containing 25 mM sodium cacodylate, pH 6.5, 30% (v/v) PEG8000, and 200 mM (NH$_4$)$_2$SO$_4$. Crystals of CsFAOMT2 were obtained in a buffer containing 100 mM sodium citrate, pH 5.5, and 1.5 M (NH$_4$)$_2$SO$_4$. The crystals were soaked briefly in the crystal-growing buffer supplemented with 10–25% (v/v) glycerol before they were flash-frozen in liquid nitrogen. Data were collected on the beamline BL19U1 and BL17U1 at the Shanghai Synchrotron Radiation Facility and processed by HKL2000[36]. The structures were solved by molecular replacement using the structure of PFOMT (PDB code: 3C3Y) as the search model. Iterative model building and refinement were performed in COOT[37] and Phenix[38]. The data collection and refinement statistics are summarized in Supplementary Table 3.

## Protein structure homology modeling and molecular docking

The protein structure homology modeling of CsFAOMT1-L50M and CsFAOMT1-Y184R mutants and the following molecular docking of SAM and EGCG into the structures were performed using Glide software (Schrodinger−Maestro version 11.9). Before docking, preprocessing was conducted on all of the structures, including hydrogen bond network optimization, removal of water molecules, and energy minimization. Four docking conformations of EGCG in each protein structure were generated, and the conformations in agreement with the results of in vitro enzymatic assays were chosen as the study models.

## Reporting summary

Further information on research design is available in the Nature Portfolio Reporting Summary linked to this article.

## Data availability

The crystal structures have been deposited in the Protein Data Bank under codes 8GXO for CsFAOMT1-SAH and 8GXN for CsFAOMT2-SAH. Raw sequences files of RNA-seq of extreme pools for BSR-Seq analysis and the 27 accession tea plant can be found at NCBI BioProject: PRJNA856047 and PRJNA855300, respectively. Source data are provided in this paper.

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

## Acknowledgements

This work was supported, in part, by the National Natural Science Foundation of China (32271933 to J.Q.J. and 32171184 to Z.M.Z.), Fundamental Research Funds for the Central Scientific Research Institute (1610212018008 to J.Q.J.), Key Scientific and Technological Grant of Zhejiang for Breeding New Agricultural Varieties (2021C02067 to J.Q.J.), the Chinese Academy of Agricultural Sciences through the Agricultural Science and Technology Innovation Program (CAASAS-TIP-2021-TRICAAS to L.C.), the China Agriculture Research System of MOF and MARA (CARS-19 to L.C.), "The Pearl River Talent Recruitment Program" of Guangdong Province (2019QN01Y979 to Z.M.Z.), and the Natural Science Foundation of Guangdong Province (2023A1515012763 to Z.M.Z.).

## Author contributions

J.Q.J., Z.M.Z., L.C., and M.Z.Y. conceived and oversaw the project. J.Q.J., F.R.Q, Q.S.L., and M.Y.W. performed most of the experiments. H.S.H., Y.Z., and Z.C. determined the structures and performed molecular docking. K.L.H. assisted identification of gene function. J.D.C. assisted with the analysis of results. W.D.D. and L.Z. assisted with qualitative and quantitative analysis of phenolic compounds. Z.M.Z. and J.Q.J. wrote the paper.

## Competing interests

The authors declare no competing interests.
