## [Peer Review File · Nature Communications]

Characterization of two O-methyltransferases involved in the biosynthesis of O-methylated catechins in tea plantReviewer #1 (Remarks to the Author):

This manuscript reports the identification of two O-methyltransferase enzymes associated with the biosynthesis of O-methylated epigallocatechin derivatives in the tea plant. This is of interest as these methylated derivatives appear to have better bioavailability and bioactivity than their non-methylated counterparts (although I wonder whether some of the methylated EGCG found in plasma might have originated from mammalian COMT enzyme activity). This knowledge could therefore be helpful in the design or breeding of tea varieties with improved phytochemical composition, although how this would be done is not really articulated.

The authors claim, on line 87, that they have identified OMTs that are "specific" for EGCG. True, their genetic and expression data strongly suggest an involvement of these enzymes in the O-methylation of EGCG, but no biochemical data are presented to support such specificity, and plant OMTs are sometimes quite promiscuous. At least, it should be shown whether the enzymes can methylate free gallic acid, the flavonoid aglycone, related flavonoid compounds, and possibly monolignol precursors. Calling the tea OMTs FAOMTs is perhaps confusing, as the authors are proposing that it is the gallic acid substituent, not the flavonoid core, that is O-methylated. In relation to this, it is important to show a scheme for the MS2 fragmentation of the enzyme products so the reader can be convinced that the methylation is indeed not on a flavonoid hydroxyl group. Finally, the kinetic data suggest that these enzymes are relatively inefficient (eg in comparison to plant caffeic acid 3-O-methyltransferase), so the specificity question definitely needs more attention.

It would have been very helpful if the authors had examined the flavanol composition and expression of the OMTs in tea roots- their experiments were restricted to the aerial parts. Tea roots have been reported to contain proanthocyanidins and, if the methylated EGCGs were present, tea hairy roots (which have been known for many years) could have been used for genetic loss of function experiments to provide unequivocal evidence for the role of the two OMTs in the biosynthesis of methylated EGCGs. As it stands, the experiment reported in the paper using injected EGCG in tobacco does not prove any functional role beyond what could be concluded from the in vitro biochemical assays. The reviewer understands that whole plant transformation in *C. sinensis* presents challenges so a hairy root approach would be useful if the compounds are found in these tissues.

The manuscript requires some attention to English and should be carefully checked by a native English speaker. In addition, there are several places where the descriptions of chemical entities are confusing. For example, the authors use the term "catechin" as describing the class of compound to which EGCG belongs. These flavanols are collectively termed "catechins", although catechin itself refers to a specific structure, distinct from that its isomer epicatechin. On line 53, use of the term "benzene ring" is inappropriate- should instead refer to the gallic acid substituent. The authors should replace the term "gene expression levels" with "transcript levels" to better describe what they are measuring.

Reviewer #2 (Remarks to the Author):

The manuscript describes the isolation of two genes from tea cultivars whose expression patterns correlate with 3" and 4" O-methylation of the natural product epigallocatechin gallate (EGCG) and present structures for the two putative O-methyltransferases (OMTs). EGCG exhibits promising bioactivities, and O-methylation has been reported to enhance oral bioavailability and improve pharmacokinetics. An understanding of the biological mechanisms of O-methylation could therefore be used to engineer tea

cultivars that yield health promoting products, so that this work in principle would be suited to publication in Nature Communications.

Having said this, there are several aspects that in my opinion preclude publication in its present form:

1. EGCG possesses two gallate-derived rings (B and D) that could potentially be methylated. It is by no means clear (to me) that the enzymes concerned are specific for only one ring. While this in principle could be detected according to MS fragmentation patterns, this are not well described. The authors refer to reference [10] (Jin et al., 2022) for identification of the products, but this does not appear to be described here either. As this is central to the conclusions being drawn, the methods and fragmentation patterns should be given here in detail.

2. Common to catechol OMTs is the methylation of vicinal hydroxyl groups. Even if the chosen enzymes are specific for the EGCG D-ring, it is not inconceivable that dimethylated and trimethylated products are generated. Similarly, I miss the action on other possible substrates such as gallic acid, methyl gallate, galloyl-CoA, hydroxycinnamates or 3,4,5-trihydroxycinnamates.

3. Only one other OMT structure (PFOMT, Kopycki et al., 2008a [25]) is mentioned at all, which was used for molecular replacement. The authors should discuss their structures in the light of other related enzymes, especially those with bound substrates / products such as SynOMT (Kopycki et al., 2008b), MxSafC (Siegrist et al., 2017), Rv0187 (Lee et al., 2019) and HsCOMT (Czarnota et al., 2019).

4. Common to the structure referred to above, the active site magnesium ion is coordinated by two vicinal hydroxyl groups, facilitating deprotonation of one of them for subsequent methylation. None of the docked EGCG structures (Figure 6) demonstrate this expected feature, calling into question the validity of the docking procedure. Returning to my second point, the authors might be able to determine the substrate binding mode of the ring to be methylated using smaller analogs.

We thank the reviewers for their critical comments on our work. We tried our best to address all their concerns and revised the manuscript accordingly. Please find our point-to-point response to each of the reviewers' comments below.

REVIEWER COMMENTS

Reviewer #1 (Remarks to the Author):

This manuscript reports the identification of two O-methyltransferase enzymes associated with the biosynthesis of O-methylated epigallocatechin derivatives in the tea plant. This is of interest as these methylated derivatives appear to have better bioavailability and bioactivity than their non-methylated counterparts (although I wonder whether some of the methylated EGCG found in plasma might have originated from mammalian COMT enzyme activity). This knowledge could therefore be helpful in the design or breeding of tea varieties with improved phytochemical composition, although how this would be done is not really articulated.

Response: We thank the reviewer sincerely for the positive comments on our work. A small paragraph has been added to the Discussion section to illustrate the potential use of the two enzymes identified in this paper for the design or breeding of tea varieties with improved phytochemical composition: By aggregating elite alleles of CsFAOMT1 and CsFAOMT2 with high transcript levels, it is possible to innovate elite tea germplasms with higher levels of O-methylated catechins in the future. Our study also identified key residues in CsFAOMT1 and CsFAOMT2. After the genetic transformation technology is mature in tea plants, these information will enable rational modification on functionally redundant O-methyltransferases to breed new tea cultivars with high O-methylated EGCG contents.

The authors claim, on line 87, that they have identified OMTs that are "specific" for EGCG. True, their genetic and expression data strongly suggest an involvement of these enzymes in the O-methylation of EGCG, but no biochemical data are presented to support such specificity, and plant OMTs are sometimes quite promiscuous. At least, it should be shown whether the enzymes can methylate free gallic acid, the flavonoid aglycone, related flavonoid compounds, and possibly monolignol precursors.

Response: Following the reviewer's suggestion, we have revised manuscript to indicate that CsFAOMT1 has a specific O-methyltransferase activity at the 3'' position of EGCG, and the EGCG3''Me level in different tissues and tea plant with different genetic backgrounds is closely

related to the transcript level of *CsFAOMT1*; whereas *CsFAOMT2* can methylate EGCG at the 3'' and 4'' positions, with EGCG4''Me as the dominant product.

We also performed *in vitro* enzymatic studies using a number of other potential phenolic substrates, including catechins ((-)-epicatechin (EC), (-)-epicatechin gallate (ECG)), phenolic acids (gallic acid, methyl gallate and caffeic acid), flavonols (isoquercitrin (quercetin 3-*O*-glucoside) and myricetin) and anthocyanin (cyanidin 3-*O*-galactoside). *CsFAOMT1* catalyzes the 3 *O*-methylation (gallic acid, methyl gallate, and caffeic acid), 3'*O*-methylation (isoquercitrin, myricetin, and cyanidin 3-*O*-galactoside), and 3''*O*-methylation (ECG) of substrates with a catechol (di-OH) ring or a pyrogallol (tri-OH) ring. However, these phenolic compounds, except for cyanidin 3-*O*-glucoside, were converted to different monomethyl products by *CsFAOMT2*.

Calling the tea OMTs FAOMTs is perhaps confusing, as the authors are proposing that it is the gallic acid substituent, not the flavonoid core, that is *O*-methylated. In relation to this, it is important to show a scheme for the MS² fragmentation of the enzyme products so the reader can be convinced that the methylation is indeed not on a flavonoid hydroxyl group. Finally, the kinetic data suggest that these enzymes are relatively inefficient (eg in comparison to plant caffeic acid 3-*O*-methyltransferase), so the specificity question definitely needs more attention

Response: A scheme for the MS² fragmentation of the methylated products had been added in Supplementary Figure 3, which support that the methylation occurs on the gallic acid substituent. We further performed *in vitro* enzymatic studies using EC as substrate, which lacks the galloyl group. But no product was detected in UPLC.

The *in vitro* enzymatic studies also revealed that *CsFAOMT1* and *CsFAOMT2* could modify isoquercitrin, myricetin, and cyanidin 3-*O*-galactoside on the flavonoid core. Phylogenetic analysis of these two proteins and *O*-methyltransferases from other species shows that they are closely related to the flavonol and anthocyanin *O*-methyltransferase (FAOMT) gene in grape. So we think it might be suitable name them *CsFAOMT1* and *CsFAOMT2*.

It would have been very helpful if the authors had examined the flavanol composition and expression of the OMTs in tea roots- their experiments were restricted to the aerial parts. Tea roots have been reported to contain proanthocyanidins and, if the methylated EGCGs were present, tea hairy roots (which have been known for many years) could have been used for genetic loss of function experiments to provide unequivocal evidence for the role of the two OMTs in the biosynthesis of methylated EGCGs. As it stands, the experiment reported in the

paper using injected EGCG in tobacco does not prove any functional role beyond what could be concluded from the in vitro biochemical assays. The reviewer understands that whole plant transformation in *C. sinensis* presents challenges so a hairy root approach would be useful if the compounds are found in these tissues.

Response: We have examined the EGCG content in tea roots (Supplementary Table 2). The presence of undetectable EGCG in the roots makes genetic experiments of loss of function using hairy roots challenging. Consistently, roots accumulated undetectable methylated EGCG, even though the transcript levels of *CsFAOMT1* and *CsFAOMT2* in roots were similar to those in flowers (Figure 2 and 4).

In the past six months, we have worked very hard to validate the activity of *CsFAOMT1* and *CsFAOMT2* in tea plants. First, we tried to transiently express these genes in the callus of tea plants. After co-culture with *Agrobacterium tumefaciens* carrying *CsFAOMT1* or *CsFAOMT2* genes, the callus turned browned after one day. Next, we spent four months working with a research group at the State Key Laboratory of Tea Tree Biology and Utilization, a leading laboratory in the field of tea research in China, to perform stable expression. However, the stable genetic transformation in tea plant presents great challenges. This system is unstable and has not yet been successful when transforming *CsFAOMT1* and *CsFAOMT2*.

We also replicated transient expression of *CsFAOMT1* in *Nicotiana benthamiana* leaves. The results again demonstrate that EGCG3''Me can only be detected when *CsFAOMT1* is expressed. We are aware that this is not the perfect experiment to verify the function of *CsFAOMT1*. However, given the complexity of tea plant cells, it may take a considerable amount of time to resolve the technical issues. We will continue our attempts on tea plants in our future studies.

The manuscript requires some attention to English and should be carefully checked by a native English speaker. In addition, there are several places where the descriptions of chemical entities are confusing. For example, the authors use the term "catechin" as describing the class of compound to which EGCG belongs. These flavanols are collectively termed "catechins", although catechin itself refers to a specific structure, distinct from that its isomer epicatechin.

Response: We thank the reviewer for his advice and have used "Catechins" as suggested. The manuscript has been checked by a professional editor to correct typos and grammatical problems.

On line 53, use of the term “benzene ring” is inappropriate- should instead refer to the gallic acid substituent.

Response: We have replaced “benzene ring” with “D-ring” and quoted Fig 1a to illustrate this.

The authors should replace the term “gene expression levels” with “transcript levels” to better describe what they are measuring.

Response: As suggested, we have replaced “gene expression levels” with “transcript levels” in the revised manuscript.

Reviewer #2 (Remarks to the Author):

An understanding of the biological mechanisms of O-methylation could therefore be used to engineer tea cultivars that yield health promoting products, so that this work in principle would be suited to publication in Nature Communications.

Response: We thank this reviewer for his positive comments on our manuscript. Our response to the reviewer's suggestions is provided below.

Comments:

EGCG possesses two gallate-derived rings (B and D) that could potentially be methylated. It is by no means clear (to me) that the enzymes concerned are specific for only one ring. While this in principle could be detected according to MS fragmentation patterns, these are not well described. The authors refer to reference [10] (Jin et al., 2022) for identification of the products, but this does not appear to be described here either. As this is central to the conclusions being drawn, the methods and fragmentation patterns should be given here in detail.

Response: Although EGCG processes two similar rings (B and D rings), our *in vitro* enzymatic assays combined with MS fragment analysis confirmed that only the D ring was modified by CsFAOMT1 and CsFAOMT2. We have added fragment patterns on the spectrograms (Supplementary Fig. 3), and described them in the revised manuscript. In UPLC, apparently only one product, which has the same retention time to the standard EGCG3''Me, was observed for CsFAOMT1. LC-MS/MS analysis further validated that the product has identical MS2 fragment ions as EGCG3''Me, of which one fragment with a m/z value of 167.09 supports the existence of a methyl group on the D-ring. For CsFAOMT2,

the UPLC and LC-MS/MS analysis support EGCG4''Me as the major product and EGCG3''Me as minor product. Trace EGCGdi-Me, which is featured by a fragment with m/z value of 181.29 on LC-MS/MS, was also observed in the reaction solution of CsFAOMT2.

To further validate the preference of CsFAOMT1 and CsFAOMT2 on the D ring of EGCG, we further performed *in vitro* enzymatic studies using EC as substrate, which lacks the galloyl group. But no product was detected in UPLC (Figure 5).

Common to catechol OMTs is the methylation of vicinal hydroxyl groups. Even if the chosen enzymes are specific for the EGCG D-ring, it is not inconceivable that dimethylated and trimethylated products are generated. Similarly, I miss the action on other possible substrates such as gallic acid, methyl gallate, galloyl-CoA, hydroxycinnamates or 3,4,5-trihydroxycinnamates.

Response: We understand the reviewer's concern. As we mentioned above, we only observed a single product, which has the same retention time to the standard EGCG3'' Me, in our *in vitro* enzymatic assay for CsFAOMT1. LC-MS/MS analysis further validated that the product has identical MS² fragment ions as EGCG3''Me, of which one fragment with a m/z value of 167.09 supports the existence of a methyl group on the D-ring. However, we could not exclude the possibility that very small amount of di-methylated and tri-methylated products are generated. For CsFAOMT2, we noticed the existence of a tiny peak in UPLC and confirmed by LC-MS/MS that the product is dimethylated EGCG.

We also performed substrate scope investigation using a number of other potential phenolic substrates, including catechins ((-)-epicatechin (EC), (-)-epicatechin gallate (ECG)), phenolic acids (gallic acid, methyl gallate and caffeic acid), flavonols (isoquercitrin (quercetin 3-O-glucoside) and myricetin) and anthocyanin (cyanidin 3-O-galactoside). CsFAOMT1 catalyzes the 3-O-methylation (gallic acid, methyl gallate, and caffeic acid), 3'O-methylation (isoquercitrin, myricetin, and cyanidin 3-O-galactoside), and 3''O-methylation (ECG) of substrates with a catechol (di-OH) ring or a pyrogallol (tri-OH) ring. However, these phenolic compounds, except for cyanidin 3-O-glucoside, were converted to different monomethyl products by CsFAOMT2.

Only one other OMT structure (PFOMT, Kopycki et al., 2008a [25]) is mentioned at all, which was used for molecular replacement. The authors should discuss their structures in the light of other related enzymes, especially those with bound substrates / products such as SynOMT (Kopycki et al., 2008b), MxSafC (Siegrist et al., 2017), Rv0187 (Lee et al., 2019) and HsCOMT (Czarnota et al., 2019).

Response: Following the reviewer's suggestion, we compared the structures of CsFAOMT1 with with that of other cation-dependent OMTs from plants, bacteria and human, including CCoACOMT from *Medicago sativa* (MsCCoACOMT, PDB code: 1SUI), PFAOMT from *Mesembryanthemum crystallinum* (PDB code: 3C3Y), CCoACOMT from *Sorghum bicolor* (PDB code: 5KVA), CCoACOMT from *Synechocytis sp.* PCC 6803 (PDB code 3CBG), CCoACOMT from *Synechocytis sp.* PCC 6803 (PDB code: 3CBG) and COMTs from human (S-CMOT, PDB code: 6I3D), *Myxococcus xanthus* (SafC, PDB code: 5LOG) and *Mycobacterium tuberculosis* H37Rv (Rv0187, PDB code: 6JCL).

It appears that CsFAOMT1 share high structure similarity with three plant-derived O-methyltransferases. All of them contain an N-terminal arm that precedes the Rossmann fold, and that arm has been shown to be critical for substrate specificity. Moreover, the bent α -helix is present in all the plant O-methyltransferases and is longer than the corresponding helices in the cation-dependent O-methyltransferases, such as the Catechol O-methyltransferases (COMT) and CCoAOMT from bacteria and animals. We have added this information in the revised manuscript.

Common to the structure referred to above, the active site magnesium ion is coordinated by two vicinal hydroxyl groups, facilitating deprotonation of one of them for subsequent methylation. None of the docked EGCG structures (Figure 6) demonstrate this expected feature, calling into question the validity of the docking procedure. Returning to my second point, the authors might be able to determine the substrate binding mode of the ring to be methylated using smaller analogs.

Response: To determine the substrate binding model, we tried to crystalize CsFAOMT1 and CsFAOMT2 in the presences of high concentrations of EGCG or gallic acid, a

smaller analog of EGCG. Unfortunately, no substrate was observed in the active sites of CsFAOMT1 or CsFAOMT2, and even we noticed that the crystals turned brown when co-crystalized with gallic acid.

We thank the reviewer for pointing out the errors in the molecular docking studies. Somehow, the electron density of Mg^{2+} was poor in the structures we used for docking. We have improved the quality of the structures, revised the coordination of Mg^{2+} , and performed the docking again. The results are consistent with the reviewer's statement that the magnesium ion in the active site is coordinated by two adjacent hydroxyl groups. CsFAOMT1-Y184 forms a hydrogen bond with the carbonyl on the galloyl group, thereby positioning the 3''-hydroxyl close to the methyl group on SAM. Substitution of the tyrosine with an arginine pushes the D-ring of EGCG deeper into the pocket, resulting in the replacement of 3''-hydroxyl group by 4''-hydroxyl group close to the methyl group on SAM.

Reviewer #1 (Remarks to the Author):

The authors have done a good job in addressing my original concerns. I better understand why it is not easy to perform loss of function assays in tea plants.

This manuscript will be of interest to researchers in plant specialized metabolism, and potentially also to the "nutraceutical" field.

There were still one or two minor grammatical errors:

Line 168. I would delete the word "apparently", unless you have questions about sensitivity of the assay such that other products could be present at lower amounts.

Line 323. should be "...this information will enable rational modification of O-methyltransferases to breed....."

Reviewer #2 (Remarks to the Author):

The authors have responded to the referees remarks with copious new data that strengthen their manuscript significantly. Nevertheless, there are a few issues remaining that should be addressed prior to acceptance to Nature Communications:

(1) I find it interesting that CsFAOMT2 action on EGCG results in a dimethylated product, attributed to a 3",4" dimethyl EGCG. What evidence is there that this is indeed the product and not 3",5" dimethyl EGCG? Both corresponding products are found using myricetin as substrate (Supplementary Figure 4f), which is reminiscent of SynOMT action on 3,4,5-trihydroxycinnamic acid (Kopycki et al., 2008b, doi 10.1074/jbc.M801943200). In this respect, CsFAOMT1 (ortho-methylation) and CsFAOMT2 (para- and ortho-methylation) are reminiscent of PFOMT (Kopycki et al., 2008a, reference 26) and SynOMT respectively. Comparisons of all four structures might shed further light on this aspect. By analogy, CsFAOMT1 should also be capable of forming 3",5" dimethyl EGCG (or syringetin from myricetin) - is this product not possible (does the structure provide clues?), or might it be present below the detection limits used? Are there e.g. steric restrictions in the CsFAOMT1 active site that preclude binding of mono-methylated EGCG that are alleviated in CsFAOMT2?

(2) The authors should provide a short discussion on why the EGCG D-ring is methylated and not the B-ring; after all, the susceptible ring in myricetin is ostensibly very similar to the B-ring. I would assume it is due to the extra stereo center in the EGCG B-ring, but this is not mentioned.

(3) It is nice that the CsFAOMT1-Y184R mutant provides a partial switch from 3" methylation specificity to 4" methylation; what about the corresponding CsFAOMT2-R184Y mutant (possibly in conjunction with M50L) - is 4" methylation reduced? This would provide important confirmation as to the relevance of these residues for their respective selectivities.

(4) The structural figures should be improved. Care should be made to inform the reader of changes in orientation (e.g. turned xxx degrees about the γ -axis from "a reference orientation" e.g. the dimer representation in Supplementary Figure 8). For this work, a simple cartoon representation with SAH, Mg²⁺ and two residues (Figure 7) is insufficient. A semi-transparent surface of the protein would be helpful to see how the substrate(s) fit to the active site. What interactions (if any) hold the A/C- and B-rings in place? What adjustments need to be made in the CsFAOMT2 active site (Figure 7d) to afford the observed 3" methylation? The issues raised above (1) concerning dimethylation should also be taken into account.

(5) Supplementary Figure 1 should be expanded (or an extra figure added) to include a structure-based alignment with the other OMTs discussed in the manuscript; I would expect a significant delineation between those methylating CoA-conjugates and others.

(6) Supplementary Figure 4 and 5 could be combined.

(7) Not so minor comment: k_{cat} should be used throughout rather than K_{cat}

(8) Prior to publishing, the manuscript would still need careful language editing.

Response to Reviewer 1

The authors have done a good job in addressing my original concerns. I better understand why it is not easy to perform loss of function assays in tea plants.

This manuscript will be of interest to researchers in plant specialized metabolism, and potentially also to the "nutraceutical" field.

There were still one or two minor grammatical errors:

Line 168. I would delete the word "apparently", unless you have questions about sensitivity of the assay such that other products could be present at lower amounts.

Line 323. should be "...this information will enable rational modification of O-methyltransferases to breed....."

Response: We thank the reviewer sincerely for the positive comments and have revised the manuscript as suggested.

Response to Reviewer 2

(1) I find it interesting that CsFAOMT2 action on EGCG results in a dimethylated product, attributed to a 3",4" dimethyl EGCG. What evidence is there that this is indeed the product and not 3",5" dimethyl EGCG? Both corresponding products are found using myricetin as substrate (Supplementary Figure 4f), which is reminiscent of SynOMT action on 3,4,5-trihydroxycinnamic acid (Kopycki et al., 2008b, doi 10.1074/jbc.M801943200). In this respect, CsFAOMT1 (ortho-methylation) and CsFAOMT2 (para- and ortho-methylation) are reminiscent of PFOMT (Kopycki et al., 2008a, reference 26) and SynOMT respectively. Comparisons of all four structures might shed further light on this aspect. By analogy, CsFAOMT1 should also be capable of forming 3",5" dimethyl EGCG (or syringetin from myricetin) - is this product not possible (does the structure provide clues?), or might it be present below the detection limits used? Are there e.g. steric restrictions in the CsFAOMT1 active site that preclude binding of mono-methylated EGCG that are alleviated in CsFAOMT2?

Response: Thank you very much for pointing out the error. It is indeed inappropriate to directly attribute the minor product to 3",4"-dimethyl EGCG. What we actually wanted to show in Supplementary figure 4d was that the product was di-methylated EGCG on the D-ring. Upon closer inspection of the LC-MS results, we identified another product that is also dimethyl EGCG on the D-ring. However, due to the lack of standard samples of 3",4"-dimethyl EGCG and

3",5" dimethyl EGCG, we were unable to differentiate them. Therefore, we labeled these two products with EGCGdi-Me in the revised figures as EGCGdi-Me1 and EGCGdi-Me2.

Following the reviewer's suggestion, we also compared the active sites of CsFAOMT1, CsFAOMT2, McPFOMT and SynOMT (as shown below). It seems that the equivalent residue of CsFAOMT1-Y184 plays critical role in determining the O-methylation site. Like the replacement of Y184 in CsFAOMT1 with R184 in CsFAOMT2, G185 in McPFOMT is substituted by H174 which contains a large side chain and may push the substrate closer to the Mg²⁺ in SynOMT.

Figure 1. The active sites of CsFAOMT1, CsFAOMT2, McPFOMT and SynOMT

Interestingly, in all the enzymatic studies of CsFAOMT1, we did not observe any dimethyl EGCG as minor product, which is different from PFOMT. By comparing the structures of CsFAOMT1 and CsFAOMT2, we found the hydrophilic residue CsFAOMT1-E49 that is in the proximity of the 5"-position of EGCG and might be involved in the recognition of 5"-OH through water-mediated hydrogen bonds, explaining why only EGCG3"Me was generated by CsFAOMT1. However, in CsFAOMT2 this residue is mutated to a hydrophobic valine (V49), which makes it possible to accommodate 5"-OMe and leads to the production of dimethyl EGCG.

We have added this information in the revised manuscript.

(2) The authors should provide a short discussion on why the EGCG D-ring is methylated and not the B-ring; after all, the susceptible ring in myricetin is ostensibly

very similar to the B-ring. I would assume it is due to the extra stereo center in the EGCG B-ring, but this is not mentioned.

Response: In our docking models, the carboxyl group next by the D-ring contributes a hydrogen bond to Y184 or Y184R, which may play critical role in the D-ring recognition and explain the specificity of CsFAOMTs to the D-ring. We have added this information in the revised manuscript.

(3) It is nice that the CsFAOMT1-Y184R mutant provides a partial switch from 3" methylation specificity to 4" methylation; what about the corresponding CsFAOMT2-R184Y mutant (possibly in conjunction with M50L) - is 4" methylation reduced? This would provide important confirmation as to the relevance of these residues for their respective selectivities.

Response: The reviewer's suggestion has been well taken. We expressed CsFAOMT2 mutants and performed *in vitro* enzymatic assay to test their activities on EGCG. Consistent to our previous finding on the key role of CsFAOMT1-Y184 for the 3"-position specificity of CsFAOMT1 on EGCG, CsFAOMT2-R184Y led to significant increase of 3"-O-methylation activity and moderate decrease of the ability to methylate EGCG on the 4"-position, and the 3"- and 4"-O-methylation activities of CsFAOMT2-R184Y could be decreased by 58% and 54% when M50L mutation was combined in CsFAOMT2, respectively (Fig. 7b).

(4) The structural figures should be improved. Care should be made to inform the reader of changes in orientation (e.g. turned xxx degrees about the y-axis from "a reference orientation" e.g. the dimer representation in Supplementary Figure 8). For this work, a simple cartoon representation with SAH, Mg²⁺ and two residues (Figure 7) is insufficient. A semi-transparent surface of the protein would be helpful to see how the substrate(s) fit to the active site. What interactions (if any) hold the A/C- and B-rings in place? What adjustments need to be made in the CsFAOMT2 active site (Figure 7d) to afford the observed 3" methylation? The issues raised above (1) concerning dimethylation should also be taken into account.

Response: In Supplementary Figure 8a and 8b (now 7a and 7b), we showed the crystal structures of the CsFAOMT1 dimer (8a) and CsFAOMT2 dimer (8b). The two structures are overlapped first, and then exhibited separately to ensure they are in the same orientation. No rotations were done in this figure.

Following the reviewer's suggestion, we provide a semi-transparent surface of CsFAOMT1 to show how EGCG fits to the active site (Supplementary figure 10). EGCG is embedded in narrow and deep cleft formed by two short α -helices, α 3 and the bent α -helix α 9, with both the A-ring and B-ring anchored through hydrogen bonds. Briefly, Y46 from α 3 interacts with the 7-OH on the A-ring. For

the B-ring, K156 and two residues from $\alpha 9$, R197 and S201, recognize 3', 4' and 5'-OH, respectively.

As for the 3''-O-methylation activity of CsFAOMT2, it might be because that the side chains of arginine are known to be more flexible than those of tyrosine, meaning that EGCG may bind in a more dynamic manner at the active site of CsFAOMT2 and CsFAOMT1-Y184R mutant, thus conferring them 3'' and 4''-methylation activity.

We have added this information in the revised manuscript.

(5) Supplementary Figure 1 should be expanded (or an extra figure added) to include a structure-based alignment with the other OMTs discussed in the manuscript; I would expect a significant delineation between those methylating CoA-conjugates and others.

Response: A Supplementary Figure (Supplementary Figure 9) including a structure-based alignment of CsFAOMTs with other OMTs discussed in the manuscript has been added in the revision. Sequence alignment also supports a closer relationship between CsFAOMTs and the plant-derived O-methyltransferases, with sequence identities of >53%, much higher than the <38% between CsFAOMTs and other OMTs from bacteria and animals.

(6) Supplementary Figure 4 and 5 could be combined.

Response: We have merged the two Supplementary Figures about the optimal enzymatic conditions of CsFAOMT1 and CsFAOMT2.

(7) Not so minor comment: k_{cat} should be used throughout rather than K_{cat}

Response: We have revised the manuscript as required.

(8) Prior to publishing, the manuscript would still need careful language editing.

Response: As suggested, the manuscript has been edited by a native English-speaker.

Reviewer #2 (Remarks to the Author):

The authors have answered my previous concerns in exemplary fashion, I congratulate them to this excellent piece of work! There are only a few very minor typographical errors, as I've noticed them I'll put them here:

line 80: contratics => contradicts

line 249: crystalized => crystallized

line 325: remove "Actually,"

line 396: change 0.1000 g to 0l.1 g or 1000 mg (?)

line 707: remove (first) "into"

Response to Reviewer 2

Reviewer #2 (Remarks to the Author):

The authors have answered my previous concerns in exemplary fashion; I congratulate them to this excellent piece of work! There are only a few very minor typographical errors, as I've noticed them I'll put them here:

line 80: contratics => contradicts

line 249: crystalized => crystallized

line 325: remove "Actually,"

line 396: change 0.1000 g to 0l.1 g or 1000 mg (?)

line 707: remove (first) "into"

Response: We thank the reviewer sincerely for the positive comments on our work and have revised the manuscript as suggested.